# Energy recuperation of driven colloids in non-Markovian baths

Félix Ginot ✉ & Clemens Bechinger ✉

When objects move through a classical fluid, their motion is primarily governed by viscous friction, which is irreversibly converted into heat. At microscopic scales, this energy dissipation presents challenges for applications, such as microscopic heat engines and microrobots relying on externally driven or self-propelled colloidal particles. In this study, we experimentally demonstrate energy recuperation (ER) in a colloidal particle driven through a viscoelastic fluid, recovering up to 30% of the energy injected into the surrounding medium as useful work. This effect, which significantly reduces the friction experienced by the particle, arises from the time-delayed response of the bath to external forces, preventing immediate relaxation to equilibrium. As a result, energy is temporarily stored, enabling bidirectional energy exchange between the non-equilibrium bath and the particle. Our experimental results are in excellent agreement with a micro-mechanical model that captures this delayed response, suggesting that similar energy recovery mechanisms could be applicable to a broad range of non-Markovian environments, including critical fluids and active baths.

Friction fundamentally governs how energy is exchanged between a particle and its surrounding medium. In Newtonian fluids, frictional forces immediately dissipate energy as heat, leading to irreversible losses. For micron-sized colloidal particles, viscous friction is so strong that motion ceases almost instantaneously once external forcing is removed[1]. This behavior is characteristic of Markovian dynamics, where the system's evolution depends only on its current state, with no memory of past interactions.

However, many real-world environments deviate from this idealized Markovian picture and instead exhibit a time-delayed response to external forces[2]. In such cases, frictional forces do not immediately dissipate energy as heat but instead cause the surrounding bath to be temporarily driven out of equilibrium, significantly altering energy exchanges[3–5]. This behavior can arise from hidden environmental degrees of freedom[6,7], complex particle-medium interactions[8–10], or external feedback mechanisms[11]. As a result, memory effects are ubiquitous across physical systems, including colloidal suspensions and glasses[12,13], critical mixtures[14,15], active matter[16,17], polymers[18,19], and biological fluids[20,21].

A key consequence of memory effects is thus that energy can be temporarily stored within the bath rather than dissipating immediately. This effect is particularly pronounced in viscoelastic media, where the storage and loss moduli can be measured using microrheology. In such systems, this delayed energy relaxation has been linked to enhanced transport properties[22–24], accelerated barrier crossing[25–27], and higher efficiency in micro-engines[28,29]. However, its potential to recover dissipated energy as useful work remains largely unexplored.

In this work, we systematically investigate how non-Markovian baths facilitate energy recuperation (ER) in a driven colloidal system. Using stochastic thermodynamics, we study energy dissipation and recovery for a particle in a harmonic trap inside a viscoelastic fluid, which acts as a model non-Markovian bath. We show that a significant fraction (30%) of the transferred energy can be recuperated as useful work when an optimal periodic driving protocol is applied. Crucially, ER is maximized when the forcing timescale matches the relaxation time of the bath. To validate our findings, we develop a minimal micromechanical model, confirming that energy recovery is a generic feature of non-Markovian environments, and extends broadly to soft matter and biological systems with slow relaxation processes. These findings open new possibilities for reducing dissipation in microscopic

Fachbereich Physik, Universität Konstanz, Konstanz, Germany. ✉e-mail: felix.ginot@uni-konstanz.de; clemens.bechinger@uni-konstanz.de

systems, including heat engines[30,31], microrobotics[32], microfluidic devices[33], and targeted drug delivery[34,35].

## Results

### Experiments

Our experiments are performed using silica particles with a diameter of 2.73 μm contained in a 100 μm-thick capillary containing a viscoelastic fluid (as described below), which serves as a model non-Markovian bath. The particle is confined within a three-dimensional optical trap by a highly focused laser beam (100× oil immersion objective) with wavelength 1064 nm. The optical potential is well described by $V(x, \lambda) = \frac{1}{2}\kappa(x - \lambda)^2$, as confirmed by the probability distribution of the particle for a static trap. Here, $x$ and $\lambda$ denote the positions of the particle and trap center, and $\kappa$ the trap stiffness, respectively. To avoid the possible influence of surface interactions, the trap center was located near the mid-plane of the capillary. For the application of periodic forces to the particle, the trap position $\lambda(t)$ was varied according to a protocol described below using a piezo-actuated stage. Particle positions were recorded with a video camera with a frame rate of 100 Hz and a spatial resolution of ±6 nm.

To demonstrate the generic character of our findings, we have used three classes of fluids which exhibit non-Markovian dynamics but strongly differ regarding the mechanism and time scale of relaxation. (i) a 8 mM aqueous solution of cetylpyridinium chloride monohydrate (CPyCl) and sodium salicylate (NaSal), which form a highly dynamic (living) network of giant worm-like micelles (WLM)[36,37]. The corresponding relaxation time has been determined via recoil experiments[38,39] to $\tau \approx 6$ s (for details, we refer to the Methods). (ii) a semidilute (0.63 wt%) solution of polyacrylamide (PAAM) polymers whose relaxation is governed by a rather complex reptation mechanism[40,41]. For the PAAM concentration considered here, the dominant relaxation time is $\tau \approx 2$ s. (iii) a suspension (10 g L⁻¹) of rather rigid tobacco mosaic viruses (TMV). Due to steric interactions between the rigid rod-like capsids[42], the system presents time-delayed dynamics with a relaxation time of $\tau \approx 0.2$ s. For comparison, we also performed experiments in entirely viscous solutions of water-glycerol (50:50) mixtures. The relaxation time of such molecular fluids is far below our experimental timescale, i.e., the fluid is not excited out of equilibrium by a driven particle. In the main text we mostly discuss the results obtained from the micellar solution, but all three systems present qualitatively similar results. All fluids were kept at 25°C. For more information regarding sample preparation and setup, we refer to the methods.

### Driving cycle and observables

The upper part of Fig. 1a, b schematically describes the driving protocol applied in our experiments. It consists of four steps and is quantified by the protocol parameter $\lambda(t)$ characterizing the time-dependent trap position. ① Starting from a fully equilibrated state, we first move the trap to the right with constant velocity $\dot{\lambda} = 0.19\,\mu\text{m s}^{-1}$ during a time interval $t_{\text{d}} = 3$ s (blue line in the main graph). Due to viscous friction, the mean particle position ($\langle x(t) \rangle$, red line) is slightly lagging behind the trap, and the particle goes up in the potential. During ②, the trap motion is paused for a time $t_{\text{neq}}$ while the particle starts to relax towards the resting trap center. For the conditions shown in Fig. 1c, $t_{\text{neq}}$ was chosen sufficiently small to prevent full relaxation of the particles, as clearly seen by the displacement between $\lambda$ and $\langle x(t) \rangle$ at the end of the pausing period. Note that experimentally there is a minimum value of $t_{\text{neq}} > 0.1$ s due to the finite response time of the stage. Eventually, during (step ③) the trap is shifted back with opposite velocity $\dot{\lambda} = -0.19\,\mu\text{m s}^{-1}$ to its initial position. Due to the incomplete relaxation during step ②, the averaged particle trajectories are not symmetric during steps ① and ③. In the final step ④, the system is allowed to fully return to thermal equilibrium during time $t_{\text{eq}} = 50$ s. To obtain sufficient statistics when computing averages, the above cycle is typically repeated at least 100 times for each data point.

Within the framework of stochastic thermodynamics, the work performed on the particle during the cycle is given by[43]

$$W[x(t)] = \int_0^t \frac{\partial V}{\partial \lambda} \dot{\lambda}\, \text{d}t = -\int_0^t \kappa(x - \lambda)\dot{\lambda}\, \text{d}t \qquad (1)$$

According to the first law of thermodynamics $\delta W = \text{d}U + \delta Q$, work is associated with changes of the inner energy $\text{d}U$ and heat $\delta Q$ dissipated within the bath. In case of a viscous (Newtonian) fluid with rapidly decaying excitations, the only source of internal energy is the potential energy of the trap, which leads to $\text{d}U_{\text{Newtonian}} = \text{d}V$.

However, due to the presence of a non-Markovian environment, a finite amount of energy, $\text{d}U_{\text{bath}}$, can be temporarily stored in the bath, which becomes transiently excited out of equilibrium. In the case of WLM, this corresponds to deformation of the filamentous network, which stores elastic energy. Other non-Markovian baths-such as glassy systems[44] or critical fluids[45]-may rely on different mechanisms for energy storage. This additional contribution supplements the potential energy of the particle in the optical trap, so that the system's internal energy is given by

$$\text{d}U = \text{d}V + \text{d}U_{\text{bath}}. \qquad (2)$$

This excess energy is slowly converted into heat as the bath relaxes to equilibrium over time $\tau_R$. Accordingly, the energy conservation reads

$$\delta W = \text{d}V + \text{d}U_{\text{bath}} + \delta Q. \qquad (3)$$

Because $U_{\text{bath}}$ is stored in the hidden, i.e., experimentally not accessible, degrees of freedom of the bath, it is by definition not accessible experimentally. However, $\delta W$ and $\text{d}V$ can be measured, which enables us to determine the total energy exchanged with the bath:

$$\delta E = \text{d}U_{\text{bath}} + \delta Q = \delta W - \text{d}V. \qquad (4)$$

As shown further below, a distinction of the two contributions of $E$ within the studied non-Markovian baths can be obtained using a simple micromechanical model.

Figure 1 c shows the averages of time-dependent work $\langle W \rangle$ (blue), potential energy $\langle V \rangle$ (orange), and exchanged energy with the bath $\langle E \rangle$ (red) for $t_{\text{neq}} = 1$ s. Because the cycle starts at equilibrium, $\langle W(0) \rangle = \langle E(0) \rangle = 0$ and $\langle V(t = 0) \rangle = \frac{1}{2}k_{\text{b}}T$. During step ①, where the particle is displaced from the center of the trap due to viscous friction, $\langle V \rangle$ increases monotonically. As the optical force acts along the direction of the trap motion, work is performed and $\langle W \rangle$ also increases. In addition, the particle injects energy into the bath while moving through the fluid, leading to an increase of $\langle E \rangle$. As the system starts from equilibrium, the corresponding energy changes during step ① are denoted as $\Delta W_{\text{eq}}$, $\Delta V_{\text{eq}}$, and $\Delta E_{\text{eq}}$, respectively. During step ②, the trap is at rest and $\dot{\lambda} = 0$. No further work is performed and $\langle W \rangle$ remains constant (see Eq. (3)). The particle relaxation within the trap reduces $\langle V \rangle$ but $\langle E \rangle$ continues to increase due to the particle motion relative to the fluid. Up to this point, the behavior is comparable to the situation in an instantaneously relaxing bath. When the trap velocity is reversed during ③, the influence of memory becomes evident, as the energy transferred to the bath, $\langle E \rangle$, exhibits a pronounced non-monotonic behavior. At first glance, this could suggest a temporary decrease in system entropy. However, this interpretation is excluded by the use of a harmonic potential, which ensures constant entropy throughout the motion. This is confirmed by comparison with a purely viscous, memoryless fluid: in that case, all applied work is dissipated as heat, and $\langle E \rangle$ increases monotonically over time (see Fig. 4a in "Methods"). Clearly, the energy exchange between the particle and the viscoelastic

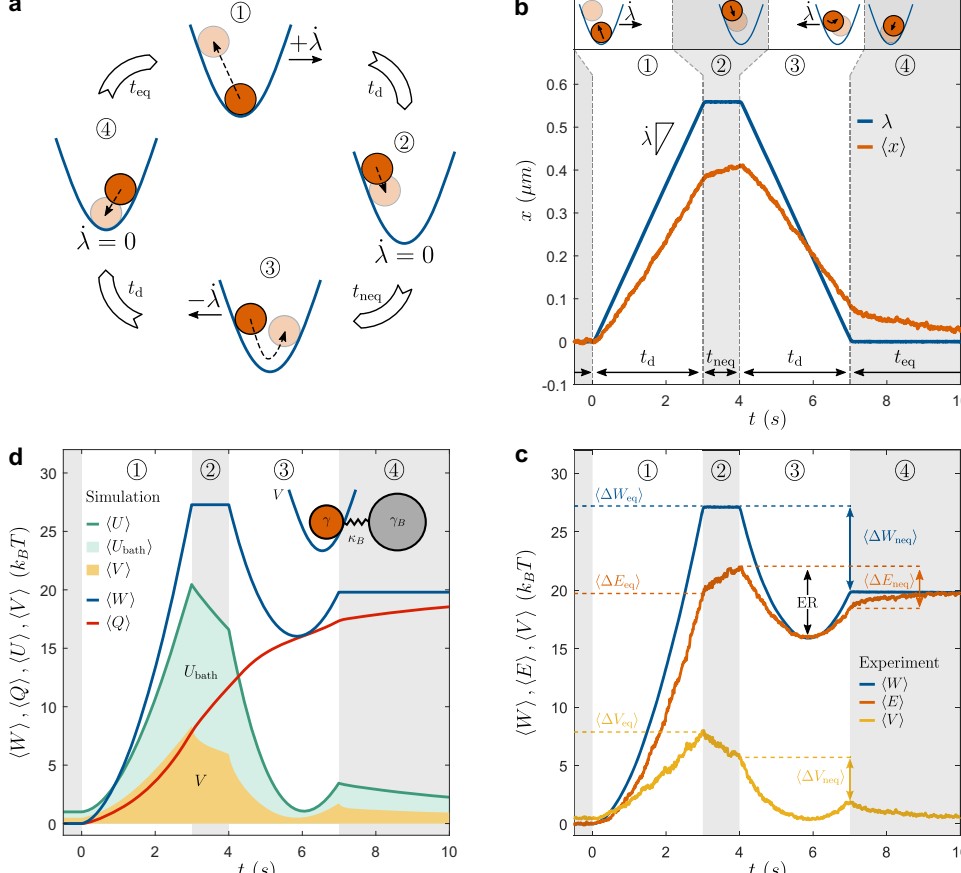

**Fig. 1 | Experimental protocol and observables. a** Sketch of the periodic driving protocol. ① Starting from equilibrium, the trap moves at constant speed to the right $\lambda$ during time $t_d$. Due to viscous forces, the motion of the trap causes the displacement of the initial position of the particle (red) in the center of the trap to the left wing (faded red). ② Pausing the trap motion for a time $t_{neq}$ causes partial relaxation of the particle back to the center of the trap. ③ Moving the trap at the same speed back to the left leads to an asymmetric particle displacement compared to ①. ④ Pausing the trap motion for a time $t_{eq}$ leading to full equilibrium of the system. **b** Time dependent and averaged positions of the trap $\lambda(t)$ (blue) and the

particle $\langle x(t) \rangle$ (red) during a typical experimental protocol, with $t_d = 3$ s, $t_{neq} = 1$ s, and $t_{neq} = 50$ s. **c** Time dependent average work $\langle W \rangle$ (blue), exchanged energy $\langle E \rangle$ (red) and potential energy $\langle V \rangle$ (orange) obtained from the same protocol. **d** Time dependent average work $\langle W \rangle$ (blue), heat $\langle Q \rangle$ (dark red), and internal energy $\langle U \rangle$ (dark green) obtained from the model. The green and orange areas correspond to the fraction of $U$ associated with $V$ and $U_{bath}$, respectively. Inset Micro-mechanical model of colloidal particle trapped within a viscoelastic bath, used for numerical simulations.

fluid during step ③ differs fundamentally from that in step ①, despite the fact that the trap motion is simply reversed and the mean particle velocity $\dot{x}$ remains nearly identical (Fig. 1b).

Because the heat $\langle Q \rangle$ must always increase on average, the initial decrease in $\langle E \rangle$ during step ③ must originate from a decrease of $\langle U_{bath} \rangle$. This is a clear signature of the hidden degrees of freedom of the bath, capable of both absorbing energy and releasing it back to the particle on timescales of seconds. While previous experiments have attributed energy transfer from the bath to the particle to thermal fluctuations observable at the single-particle level[46,47], a key distinction here is that the energy transfer we report persists on average over many trajectories, indicating a deterministic component arising from the non-Markovian nature of the bath. In the following, we refer to the amount of energy transferred from the bath to the system, up until $\langle E \rangle$ reaches a local minimum, as ER (see Fig. 1c). As the system starts out of thermal equilibrium, the corresponding changes in work, potential energy, heat, and exchanged energy during step ③ are denoted as $\Delta W_{neq}$, $\Delta V_{neq}$, and $\Delta E_{neq}$, respectively. Before discussing the physical origin of ER, we want to highlight that both $\langle V \rangle$ and $\langle W \rangle$ also exhibit a non-monotonic behavior, similar to $\langle E \rangle$, during step ③. This is however fully expected, and simply arises from the storage and recovery of potential energy, which can also happen in a viscous bath (see Fig. 4a in "Methods").

To ensure that our findings were not specific to a single fluid, we conducted additional experiments using tobacco mosaic virus (TMV) and PAAM polymer solutions, both of which exhibit time-delayed responses. In both cases, we consistently observed ER (see Fig. 4c, d in "Methods"). However, as expected, this effect was absent in a purely viscous fluid (see Fig. 4a, b in "Methods"), reinforcing that it arises from the bath's non-Markovian properties rather than from the particular fluid used in the main experiment.

## Microscopic model

To rationalize our experimental findings, we conducted numerical simulations based on a minimal micro-mechanical model. This model describes the coupling between the colloidal particle and the viscoelastic fluid by introducing a fictitious bath particle connected to the colloid via a harmonic spring (see Fig. 1d inset). This setup effectively captures the time-delayed response of the bath, with the spring accounting for the slow hidden degrees of freedom that can store energy. Consistent with our experimental measurements, we define the system under study as only the colloidal particle and the harmonic potential. While one could theoretically employ a Markovian embedding by including the bath particle within the system-a framework that would recover classical Markovian dynamics-this requires direct access to hidden degrees of freedom that are often impractical to

measure in complex experimental systems. Given these limitations, we argue that our approach of coarse-graining these slow-decaying hidden degrees of freedom into a non-Markovian bath provides a more practical and insightful framework. The resulting non-ideal bath is evident within the Generalized Langevin Equation (GLE) framework (see "Methods"), where the noise term is now colored. Despite its simplicity, this model is well-supported by experimental studies in various viscoelastic solutions[23,26,38,39].

Considering an additional external potential $V(x) = \frac{1}{2}\kappa(x-\lambda)^2$, the positions of the colloidal particle ($x$) and the bath particle ($x_b$) are described by the following Langevin equations:

$$\gamma \dot{x} = -\kappa_b(x - x_b) - \kappa(x - \lambda) + \xi(t) \quad (5)$$

$$\gamma_b \dot{x}_b = -\kappa_b(x_b - x) + \xi_b(t) \quad (6)$$

where $\gamma$ and $\gamma_b$ represent the friction coefficients of the colloidal and bath particles, respectively, and $\kappa_b$ is the coupling strength between them. $\xi$ and $\xi_b$ are independent, delta-correlated random forces with zero mean.

Although the two coupled Langevin equations can be reformulated as a single GLE with an exponentially decaying memory kernel[48,49], we favor the micro-mechanical model used here. In the GLE framework, viscoelasticity is encoded in a memory kernel, from which one can derive the storage and loss moduli, $G'$ and $G''$-standard tools in microrheology (see "Methods"). However, the GLE treats memory as an effective response, without revealing its physical origin. By contrast, this micro-mechanical model explicitly includes internal degrees of freedom of the bath, providing a clearer physical picture of how energy is stored and released. It also facilitates direct calculations of stochastic observables (see "Methods"), making it both more intuitive and more versatile for our study.

The system exhibits two characteristic timescales: one for the relaxation of the colloidal particle inside the trap, $\tau_t = \gamma/\kappa \approx 0.2$ s, and another for the bath particle. Since the colloidal particle remains confined within the optical potential throughout the experiment, the overall system's relaxation is dominated by the bath particle's relaxation time, given by $\tau_b = \gamma_b/\kappa_b \approx 6$ s. All material-specific parameters of the model ($\gamma$, $\gamma_b$, and $\kappa_b$) were determined by fitting the experimentally measured work, potential energy, and exchanged energy (see Fig. 5 in the "Methods" section). For the viscoelastic micellar solution studied in Fig. 1, the corresponding values are $\gamma = 0.34$ μN s m$^{-1}$, $\gamma_b = 6.33$ μN s m$^{-1}$, and $\kappa_b = 1.06$ μN m$^{-1}$ (see Table 1).

From this, we immediately obtain $\langle W \rangle$ (blue line) and $\langle V \rangle$ (orange area), as shown in Fig. 1d, which nearly perfectly match the experimental data. We can now also compute the total internal energy $\langle U \rangle$ (dark green line) and the stochastic heat $Q = W - U$ (dark red line), both of which are plotted in Fig. 1d. It is worth noting that $Q$ can also be directly calculated from the motion of the colloidal and fictitious bath particles (see "Methods"). The fraction of internal energy stored within the bath, $U_{bath} = \frac{1}{2}\kappa_b(x - x_b)^2$, is represented by the light green area.

Within our micromechanical model, the ER process is understood as follows: When the colloidal particle is driven to the right (①), the spring connecting it to the bath particle becomes stretched, increasing $U_{bath}$. During ②, when the trap is held stationary for a time $t_{neq}$, the spring begins to relax, causing a decrease in $U_{bath}$. In particular when $t_{neq} < \tau_b$, the relaxation is incomplete, explaining why $U_{bath}$ decreases only slightly during ②. Upon moving the trap back to the left in ③, the colloidal particle moves against the still-extended spring, which results in a rapid release of $U_{bath}$, part of which is effectively converted into useful work. The model also clarifies that the experimentally measured ER includes two contributions: a (negative) energy transfer from $U_{bath}$ and $V$, and a (positive) contribution from heat dissipation $Q$, generated by the particle's motion. Notably, the slope of $Q$ (the rate of heat

## Table 1 | Model parameters used for the simulations in the manuscript

| $\kappa$ (μN m$^{-1}$) | $\gamma$ (μN s m$^{-1}$) | $\gamma_b$ (μN s m$^{-1}$) | $\kappa_b$ (μN m$^{-1}$) |
|---|---|---|---|
| 1.95 | 0.34 | 6.33 | 1.06 |
| 1.28 | 0.19 | 10.6 | 1.09 |

$\kappa = 1.95$ μN m$^{-1}$ was used for Figs. 1 and 2. $\kappa = 1.28$ μN m$^{-1}$ was used for Fig. 3.

dissipation) during step ③ is substantially lower than during step ①, confirming that a significant portion of the bath's stored energy is recovered as mechanical work rather than lost as heat. As a result, although the amplitude of the particle's motion is similar in both steps, the friction experienced by the particle on the return path is significantly reduced.

## Optimization of energy recuperation

Our model suggests that ER is linked to the non-equilibrium nature of the bath, and thus heavily depends on the partial relaxation step and the duration of $t_{neq}$. Figure 2a shows the measured amount of ER during ③ (symbols) as a function of $t_{neq}/\tau_b$. Consistent with our model (solid lines), ER decreases with increasing $t_{neq}$ and approaches zero when $t_{neq} \gtrsim \tau_b$. When comparing the measured values of $\langle \Delta W_{neq} \rangle$, $\langle \Delta V_{neq} \rangle$, and $\langle \Delta E_{neq} \rangle$, good agreement is found between experiments and the model, particularly for $t_{neq}/\tau_b \leq 1$. Opposed to merely viscous baths, in our experiments $\langle \Delta W_{neq} \rangle$ is systematically lower than $\langle \Delta V_{neq} \rangle$ when $t_{neq}/\tau_b \ll 1$, confirming that additional energy can be extracted from the bath. As a side note, we mention that the agreement between experiments and the model can be significantly improved by adding a second bath particle[38], resulting in an additional relaxation time (see Fig. 5b in "Methods").

Since recuperated energy must have been previously injected into the bath, a strong correlation between the amount of ER and $\Delta E_{eq}$-the energy injected into the bath during ①-is expected. Due to energy conservation, this must hold both for averaged values but also on a single trajectory level. This is confirmed in Fig. 2b, which shows the experimental data for different values of $t_{neq}$ under the given protocol. As $t_{neq}$ increases, the system has more time to relax, resulting in a decrease in both ER and its correlation with $\Delta E_{eq}$. For very large $t_{neq}$, the previously injected energy is fully dissipated as heat, causing both ER and any correlations to vanish. Notably, for $t_{neq} = 1$ s, we typically recover ~30% of the energy that would otherwise have been lost to the bath.

Naturally, when a particle is driven in a viscoelastic fluid, some of the work is temporarily stored in the bath-a well-known consequence of the fluid's storage modulus, often illustrated using Lissajous curves[50]. However, we have shown that this stored energy can subsequently be reinjected into the particle, allowing it to perform useful work. This process of ER can help mitigate frictional losses, a common challenge for micromachines operating in fluid environments.

This process bears some resemblance with the operation of a classical heat engine, where a piston moves within a cylinder containing a compressible working gas. In such engines, energy is cyclically injected and extracted through the compression and expansion of the gas, with part of the input work stored temporarily as internal energy and the rest dissipated as heat due to friction and other losses. Similarly, in our system, the colloidal particle (piston) is periodically driven within a harmonic trap (cylinder) immersed in a viscoelastic fluid (working medium). During forward motion, the particle squeezes the viscoelastic network, analogous to compressing a gas, thereby injecting energy into both the trap and the surrounding medium. Due to the fluid's delayed response, a fraction of this energy is stored elastically in the bath's internal degrees of freedom rather than being

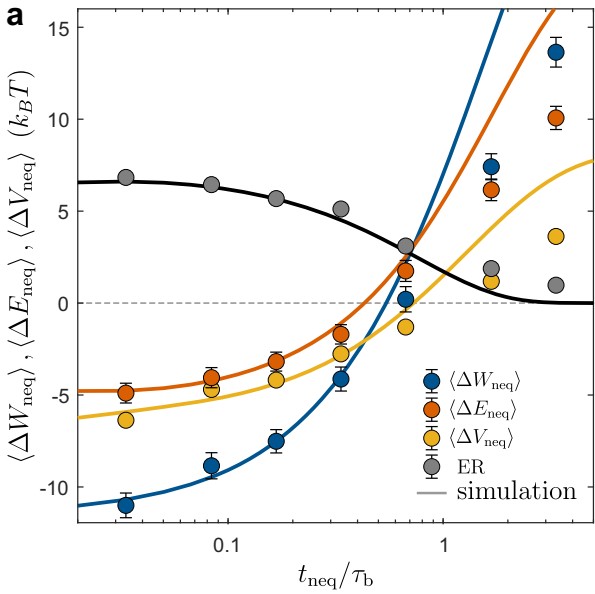

**Fig. 2 | Energy recuperation and correlations. a** $\langle\Delta W_{\mathrm{neq}}\rangle$ (blue), $\langle\Delta E_{\mathrm{neq}}\rangle$ (red), $\langle\Delta V_{\mathrm{neq}}\rangle$ (orange), and ER amplitude (black) for increasing values of $t_{\mathrm{neq}}/\tau_{\mathrm{b}}$. Simulations (lines) initially show good agreement with experiments (symbols), with systematic deviations for larger values of $t_{\mathrm{neq}}/\tau_{\mathrm{b}}$. Error bars correspond to the standard error of mean (SEM), and are not shown when smaller than symbols. **b** Correlation between $\Delta E_{\mathrm{eq}}$ and $\Delta E_{\mathrm{neq}}$ for $t_{\mathrm{neq}} = 1, 3$, and 10 s (dark to light red). Symbols correspond to experimental data, dashed-lines to simulations. The Newtonian bath doesn't exhibit any correlation (black dashed-line).

immediately dissipated. Upon reversing the trap motion, part of the stored energy is released and can be reinjected into the system, analogous to the expansion stroke of the piston where stored energy contributes to useful output. However, just as in macroscopic engines, not all energy is recovered; viscous friction leads to irreversible heat dissipation, thereby limiting the efficiency of the cycle.

As a measure of the fraction of irreversibly lost energy (heat) during a cycle, we define the work ratio $\Sigma = \Delta W_{\mathrm{neq}}/\Delta W_{\mathrm{eq}}$, which can be directly obtained from our data. In Fig. 3, we plot $\Sigma$ as a function of the driving time $t_d$, for cycles with a fixed particle displacement of $\Delta x = 0.56\,\mu\mathrm{m}$ and $t_{\mathrm{neq}} = 1\,\mathrm{s}$. With this definition and sign convention, we have $-1 < \Sigma < 1$, where the lower limit $\Sigma = -1$ represents the dissipation-free case-where the work expended during step ① is fully recovered during step ③, so that $\Delta W_{\mathrm{neq}} = -\Delta W_{\mathrm{eq}}$. The upper limit, $\Sigma = 1$, occurs when all energy spent and released during compression and expansion is dissipated as heat, thus $\Delta W_{\mathrm{neq}} = \Delta W_{\mathrm{eq}}$.

To understand the minimum in $\Sigma$, we recall that the system's internal energy, $U = V + U_{\mathrm{bath}}$, is the sum of the particle's potential energy in the optical trap ($V$) and the elastic energy stored in the bath ($U_{\mathrm{bath}}$). Both $V$ and $U_{\mathrm{bath}}$ are excited when the particle is driven by the trap, but they relax at very different rates: $\tau_{\mathrm{t}} = \gamma/\kappa \approx 0.2\,\mathrm{s}$ for $V$, and $\tau_{\mathrm{b}} = \gamma_{\mathrm{b}}/\kappa_{\mathrm{b}} \approx 6\,\mathrm{s}$ for $U_{\mathrm{bath}}$. Because of this disparity, energy is first primarily stored in the faster channel ($V$), with the slower bath channel ($U_{\mathrm{bath}}$) responding more gradually. Only when the system has reached a non-equilibrium steady state (NESS) will both channels be fully loaded. However, in our experiments, this steady state is generally not reached, and the energy partition between the two components depends sensitively on the driving time $t_d$. For short driving time $t_d$ (corresponding to higher trap velocities), most of the work is stored in $V$ (the faster channel). But since $\tau_T \ll t_{\mathrm{neq}}$, this energy rapidly dissipates as heat during step ②, resulting in a higher value of $\Sigma$. As $t_d$ increases, a larger fraction of the work begins to accumulate in $U_{\mathrm{bath}}$, the slower channel. Although the system has more time to relax in this regime, the relaxation of $U_{\mathrm{bath}}$ remains intrinsically slow, which limits heat dissipation during step ② and thereby lowers $\Sigma$. This trade-off results in a minimum of $\Sigma$ at intermediate driving velocities, where energy is preferentially stored in the slow channel but not yet entirely dissipated. For large $t_d$ values,

the resulting trap velocities get very low, and the system approaches quasi-equilibrium. Then most of the work is once again dissipated as heat, driving $\Sigma$ back toward unity.

For comparison, we include the results of a Langevin simulation for a particle undergoing the same cycle in a Newtonian liquid with the same zero-shear viscosity (dashed line). As expected, slower trap movement allows more time for relaxation, leading to a higher $\Sigma$. Notably, for a wide range of $t_d > 1\,\mathrm{s}$, $\Sigma$ in a viscoelastic bath is lower than that of a Newtonian liquid with identical zero-shear viscosity. This demonstrates that heat losses in a periodic driving protocol can be reduced by utilizing environments with additional slowly relaxing degrees of freedom.

## Discussion

In this work, we have demonstrated that a colloidal particle driven in a non-Markovian environment can recover a significant fraction of the energy it injects into the medium. The delayed response of the bath leads to a transient accumulation of internal energy, setting the medium out of equilibrium. This stored energy is then partially reinjected into the particle as useful work, effectively reducing dissipation.

While viscoelastic fluids are already known to store mechanical energy, our study establishes that this energy can be systematically recovered in a controlled manner. By studying three different viscoelastic fluids and validating our results with a micromechanical model, we confirm that ER is not an artifact of a specific material but a generic consequence of a bath's time-delayed response. This makes ER a broadly applicable mechanism, relevant to a variety of soft matter systems with memory effects.

Furthermore, we show that ER can be optimized by carefully tuning the driving protocol. Specifically, maximal energy recovery occurs when the forcing timescale matches the characteristic relaxation time of the bath. This highlights the possibility of engineering dissipation in microscopic systems by adapting external forcing to the bath's intrinsic timescales.

Beyond reducing energy losses in driven or self-propelled particles-relevant for various applications[51]-ER could play a crucial role in the design of microscopic colloidal engines. Previous experimental realizations of colloidal Stirling and Carnot engines[30,31,52] have largely avoided mean translational motion due to the strong viscous friction

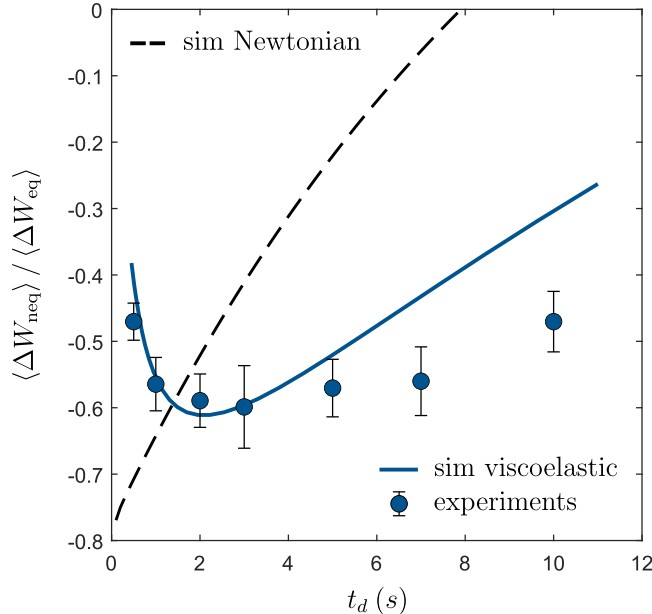

**Fig. 3 | Optimization of work ratio.** Work ratio $\langle W_{\mathrm{neq}}\rangle/\langle W_{\mathrm{eq}}\rangle$ as a function of $t_d$ for protocol parameters $t_{\mathrm{neq}} = 1\,\mathrm{s}$ and $\Delta x = 0.56\,\mu\mathrm{m}$. Symbols and full line correspond to experiments and simulations in a micellar viscoelastic solution, respectively. Error bars correspond to the standard error of mean (SEM). The dashed line corresponds to the result of a Langevin simulation of an overdamped particle in a viscous liquid with the same zero-shear viscosity (as the micellar fluid) which is subjected to the same driving protocol.

of the solvent. Instead, these studies focused on variations in particle fluctuations[53] within a stationary trap with time-dependent stiffness, immersed in a temperature-modulated heat bath. While effective, this approach limits the amount of extractable work and makes it difficult to harness mechanical energy for practical applications.

By leveraging ER, colloidal micro-engines capable of cyclic forward-backward motion-akin to a piston in a macroscopic heat engine-become feasible. This not only increases the amount of extractable work but also enables direct mechanical coupling to components such as gears and axles. More broadly, since ER arises from the bath's memory rather than specific material properties, we expect similar energy recovery effects to be relevant in other non-Markovian environments, such as critical fluids[15], active matter[54,55], and glassy systems[44,56]. These systems exhibit tunable relaxation times, offering new opportunities to control dissipation and improve energy efficiency in microscopic machines.

## Methods

### Solution and sample preparation
The principal experiments were conducted utilizing 2.73 μm silica particles procured from microParticles GmbH®, which were dispersed in a 8 mM equimolar solution comprised of CPyCl monohydrate (99%, Sigma®) and sodium salicylate (99.5%, Sigma®). The mixture was subjected to slow agitation at a rate of 100 rpm for 24 h at 45°C to ensure thorough mixing. Subsequent to this, the temperature was reduced to 25°C to facilitate sample storage while maintaining the agitation rate. Under such conditions, the fluid forms giant WLM, characterized by a tube diameter of ≈3 nm, a persistence length of ≈10 nm, and a mesh size of ≈30 nm, exhibiting pronounced viscoelastic properties[36,37]. Notably, the system can be aptly characterized by a single time-scale, determined to be ≈6 s through recoil experiments[38].

To prepare a sample, we first put 1 mL of the solution into an Eppendorf® tube and add ≈0.5 μL of a concentrated solution of 2.73 μm silica particles. Subsequently, a 100 μm thick capillary (Rectangle Boro

Tubing, VitroTubes®) is immersed in the solution, which immediately fills with the fluid due to capillary forces. The capillary is then sealed using bee wax (Hampton Research®), supplemented by a layer of two-component epoxy glue at both ends to ensure chemical inertness and mechanical stability of the seal. The capillary is affixed onto a glass slide to facilitate handling. The glass slide is engineered with a central window to allow the capillary to remain suspended. To achieve complete equilibration, samples are typically maintained 48 h on a temperature-controlled sample holder before the start of the experiment. To enhance reproducibility, the data presented in Fig. 1 and Fig. 2 were derived from the identical sample of the same solution. In contrast, the data depicted in Fig. 3, were obtained using a secondary solution.

### Particle detection
Particle illumination is achieved using a conventional Köhler configuration, incorporating a blue LED illumination source (470 nm), and a high numerical aperture (0.78 NA) condenser lens (Thorlabs® CSC2001). The collected light is transmitted through the objective, and the particle image is projected onto a CCD camera (Basler® cA2040-120uc). This system operates at 100 frames per second (fps), with a region of interest (ROI) measuring 100 × 100 pixels. Typically, the particle image manifests as a bright spot with a Gaussian profile centered against a darker background. Detection of the bright spot is conducted via a customized algorithm inspired by ref. 57. Initially, this algorithm applies a Gaussian filter with a kernel size commensurate with the particle dimension (15px) to achieve image smoothing. It then identifies the brightest pixel and fits a second-order polynomial to measure the particle position with accuracy ≈5 nm.

### Optical tweezers setup
The optical potential is generated using a 1064 nm fiber laser (IPG Photonics® YLM-5), with its intensity modulated by an acousto-optical modulator (AOM−Gooch & Housego®). The laser beam is subsequently directed to the back-aperture of a 100× oil-immersion microscope objective (Olympus® UPLxAPO100xO, NA = 1.45), which focuses the beam into the sample cell, thus creating a harmonic trap. To mitigate heating effects, the laser power is constrained at 50 mW. The samples are mounted on a piezo-actuated translational stage (Piezoconcept® LT3.300), enabling trap movement relative to the sample cell. The sample cell is affixed to a copper block to maintain a constant temperature, controlled at 25°C via a water thermostat. Additionally, the oil immersion objective, which is in direct contact with the sample cell, is maintained at the same temperature utilizing an objective heater (Okolab®). Trap stiffness $\kappa$ is measured from a static experiment, wherein the optical potential is derived from the position probability density function of a particle using the relation $V(x) = -k_{\mathrm{b}}T\log(P(x)) + \mathrm{cst} = \frac{1}{2}\kappa x^2$ (note that in the reference frame of the camera the trap position always fixed to $\lambda := 0$).

### Synchronization of the camera and the stage
To properly compute the microscopic observables (work, heat, internal energy) of our experiments, it is necessary to synchronize all the elements. Each camera frame is triggered by an external analog signal generated with a function generator (Tektronix® AFG31000). The piezo-stage motion is controlled by the same function generator, using an analog input line. This ensures that both systems are controlled by the same internal clock of the function generator and stay a priori synchronized at all times. Additionally, we use a DAQ card (National Instrument USB-6003), which records both the output exposure signal of each captured frame and the position of the piezo-driven stage at a much higher frequency (10 kHz) than the image acquisition rate. During the analysis, we have a parsing step where we use the recorded signal of the DAQ card to associate each image with a corresponding stage position. This ensures synchronization *a posteriori* and allows us to detect any issues with the recording.

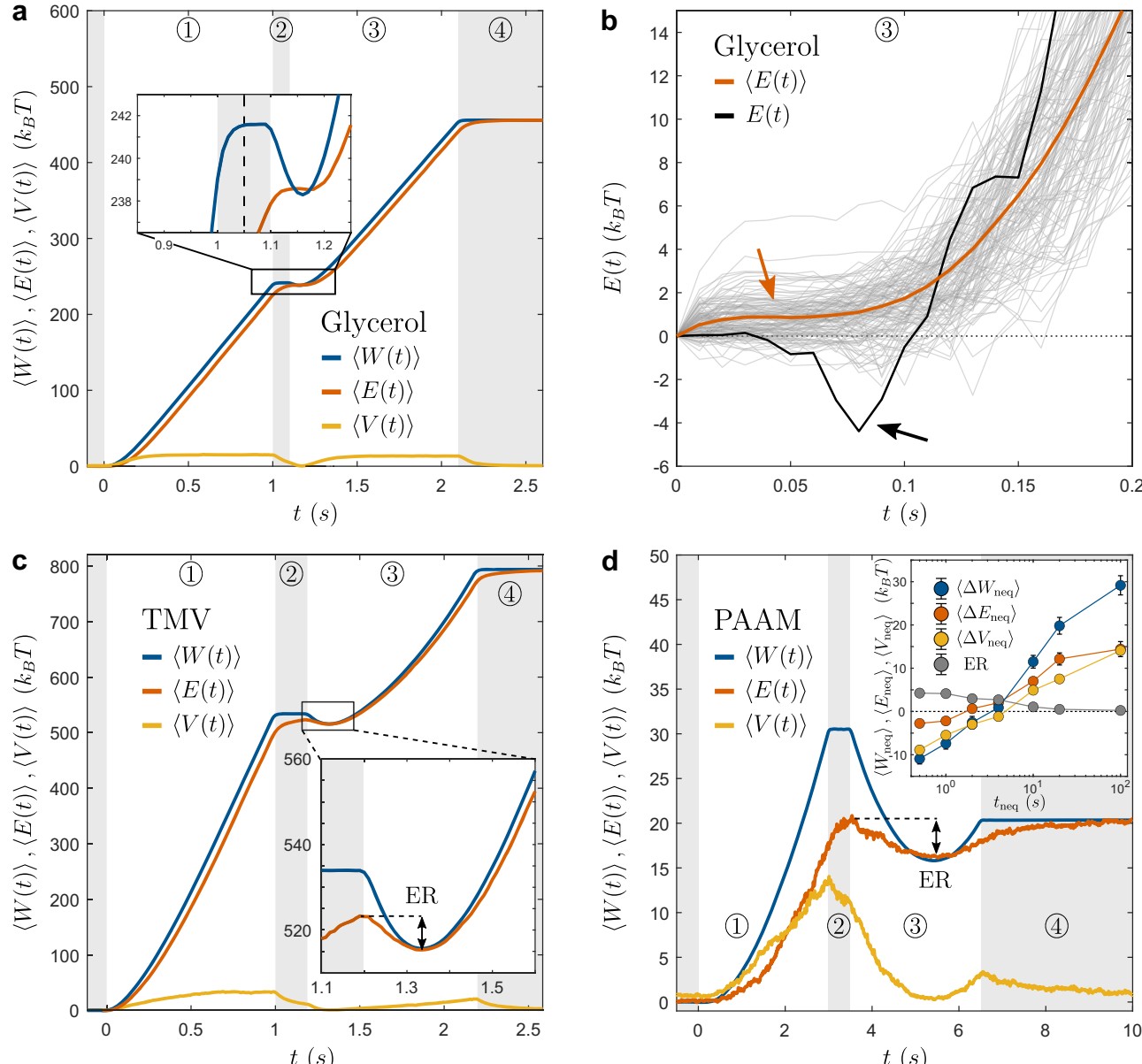

**Fig. 4 | Energy recuperation in other fluids. a** Average work $\langle W(t)\rangle$ (blue line), exchanged energy $\langle E(t)\rangle$ (red line), and potential energy $\langle V(t)\rangle$ (orange line), measured inside a 50% water-glycerol solution, which presents purely Newtonian properties. As expected, we do not observe the presence of energy Recuperation. **b** Highlight of step ③, for a 50% water-glycerol solution. For better visualization, time and E axis were shifted to zero. Faded lines correspond to some of the individual $E(t)$ trajectories used to compute the average $\langle E(t)\rangle$ (red line). Note how here $\langle E(t)\rangle$ always increases monotonically (red arrow), while for single trajectories (black line), exchanged energy increases non-monotonically and can even dip below zero (black arrow). **c** Average work $\langle W(t)\rangle$ (blue line), exchanged energy $\langle E(t)\rangle$ (red line), and potential energy $\langle V(t)\rangle$ (orange line), measured inside a 10 g L$^{-1}$ TMV solution. Although comparatively smaller than those observed in CPyCl-NaSal and PAAM solutions, the presence of ER (highlighted in the zoomed area) demonstrates it is a generic property of viscoelastic baths. **d** Average work $\langle W(t)\rangle$ (blue line), exchanged energy $\langle E(t)\rangle$ (red line), and potential energy $\langle V(t)\rangle$ (orange line), measured inside a 0.63 wt% PAAM solution. Inset: Differential work ($\langle \Delta W_{neq}\rangle$ in blue), exchanged energy ($\langle \Delta E_{neq}\rangle$ in red), potential energy ($\langle \Delta V_{neq}\rangle$ in orange), and ER amplitude (black) for increasing values of $t_{neq}$. The resulting curves exhibit a qualitative similarity to those obtained with the 8 mM CPyCl-NaSal solution in Fig. 1c and Fig. 2a.

## Protocol parameters

For experiments illustrated in Fig. 1, we used a driving protocol with $t_d = 3\,s$ and $\dot{\lambda} = 0.19\,\mu m\,s^{-1}$, while exclusively varying $t_{neq}$. Each data point is obtained from ≈100 repetitions of the protocol, and error bars are calculated using the standard error of the mean ($SEM(x) = std\,(x)/\sqrt{N(x)-1}$), where N denotes the number of individual measurements. During the protocol, the typical strain length is $l_d = 600\,nm$, which significantly exceeds any of the length-scales of the WLM. We also checked that the fluid is in the linear regime, ensuring that the fluid's properties, specifically its relaxation time, are

independent of driving speed $\dot{\lambda}$. Moreover, $t_d = 3\,s$ is comparable to the typical relaxation of the fluid $\tau_b \approx 6\,s$, which sets the system in a regime where the non-equilibrium effects are maximized. This is made clear in Fig. 3 where we show that the best ratio of work recovered after the cycle is obtained for $t_d \sim 3\,s$.

## Absence of energy recuperation in a Newtonian fluid

The presence of ER is a direct consequence of the bath's viscoelastic (non-Markovian) properties. To verify the absence of ER in a Newtonian (Markovian) fluid, we conducted similar experiments using a

**Table 2 | Protocol parameters used in the experiments**

| Solution | $t_d$ | $\lambda$ | $t_{neq}$ | $t_{eq}$ |
|---|---|---|---|---|
| - | (s) | (µm s$^{-1}$) | (s) | (s) |
| CPylCl | 3* | 0.19 | 1* | 50 |
| Water-Glycerol | 1 | 5 | 0.1 | 10 |
| PAAM | 3 | 0.13 | 0.5* | 100 |
| TMV | 1 | 2 | 0.2 | 9 |

Parameters with * are varied in some figures.

50% water-glycerol solution which leads to a viscosity $\eta \simeq 8$ mPa s. Note that this is quite lower than the zero-shear viscosity of the 8 mM CPylCl solution $\eta \simeq 300$ mPa s. However, to reach such high viscosity one needs a 90% glycerol ratio, which strongly affects the refraction index of the fluid and makes the optical trapping ineffective.

The results are shown in Fig. 4a, where we plot the average work $\langle W(t) \rangle$ (blue line), energy exchanged with the bath $\langle E(t) \rangle$ (red line), and potential energy $\langle V(t) \rangle$ (orange line). As expected, even in a Newtonian fluid, work $\langle W(t) \rangle$ can still be extracted on average-at the expense of potential energy-resulting in a non-monotonic increase. However, in contrast to non-Markovian systems, the average bath-exchanged energy $\langle E(t) \rangle$ (red line) exhibits a purely monotonic increase. This behavior is emphasized in Fig. 4b, where we zoom in on the early phase of ③ (see red arrow). We further illustrate this by showing individual trajectories of energy exchange $E(t)$ (faded lines), which contribute to the average (red line). When examining a single trajectory, such as the one highlighted in black (see black arrow), we observe that the energy exchange with the bath increases non-monotonically, even dipping below zero. This confirms that in the absence of hidden degrees of freedom, the internal energy simply matches the potential energy ($U = V$), and the energy exchanged with the bath is purely heat ($E = Q$), which, on average, must increase monotonically. All the protocol parameters are provided in Table 2. For the Markovian simulations, we employed the same Langevin model as before (Eq. (5)), but with a modified friction coefficient $\gamma_{Newtonian} = \gamma + \gamma_b$, ensuring both Newtonian and viscoelastic simulations experience identical total friction. Additionally, we set $\kappa_b = 0$, effectively disconnecting the bath particle from the colloidal particle, simplifying the Langevin equation to:

$$(\gamma + \gamma_b)\dot{x}(t) = -\nabla V + \xi(t) \qquad (7)$$

### Energy recuperation in other viscoelastic baths
To further demonstrate that our findings are not specific to certain materials, but a generic feature of viscoelastic baths, we investigated the presence of ER in two additional viscoelastic fluids. First, we conducted experiments with a semidilute 0.63 wt% polyacrylamide (PAAM) solution[40,41], which exhibits typical polymer reptation dynamics with a relaxation time of ~2 s. Second, we investigated a biological system composed of tobacco mosaic virus (TMV) at a concentration of 10 g L$^{-1}$. Due to steric interactions between the rigid, rod-like capsids of TMV[42], the suspension displays weak but noticeable viscoelastic properties, with a characteristic relaxation time of around 0.2 s. These relaxation times were measured through recoil experiments[38].

Figure 4 c and d, display the time-dependent average work ($\langle W(t) \rangle$, blue line), exchanged energy ($\langle E(t) \rangle$, red line), and potential energy ($\langle V(t) \rangle$, orange line) for the TMV and PAAM solutions, respectively. In both cases, $\langle E(t) \rangle$ exhibits a non-monotonic increase, indicating the occurrence of ER. Additionally, for the PAAM solution (see inset), we show the work ($\langle \Delta W_{neq} \rangle$ in blue), exchanged energy

($\langle \Delta E_{neq} \rangle$ in red), potential energy ($\langle \Delta V_{neq} \rangle$ in red) differentials, and ER amplitude (black), for increasing values of $t_{neq}$. Here, we recover a qualitatively similar trend to that depicted in Fig. 2a for the 8 mM CPyCl-NaSal solution. Despite the existing differences in material properties and relaxation behavior of these systems (more than one order of magnitude for the relaxation time), the presence of ER in both the PAAM solution and the TMV system indicates that this feature is generic to viscoelastic baths. The corresponding protocol parameters can be found in Table 2.

### Extracting the parameters of the microscopic model
To obtain the model parameters that match the experiments best, we use a gradient descent algorithm. We run Langevin simulations of the full cycle using Eqs. (5) and (6), for iteration $i$ we obtain curves for $W_s^i(t)$, $E_s^i(t)$, and $V_s^i(t)$. Note that for a faster process, we remove the noise, so that we only need to compute one instance of the simulation. We then compare these simulations with the experimental curves $\langle W(t) \rangle$, $\langle E(t) \rangle$, and $\langle V(t) \rangle$. For iteration $i$, we thus obtain the following mean square difference:

$$S^i = \int_0^\infty ((W_s^i - \langle W \rangle)^2 + (E_s^i - \langle E \rangle)^2 + (V_s^i - \langle V \rangle)^2)dt \qquad (8)$$

We start with an initial set of simulation parameters ($\gamma^0$, $\gamma_b^0$, $\kappa_b^0$), and compute $S^0$. We perform an initial update of all the parameters (we start by arbitrarily increasing them by 1%). We then compute intermediate mean square differences in the same way as in Eq. (8), to estimate the individual impact of each parameter variation and obtain the initial slope for the gradient descent. In the following the intermediate integrals for iteration $i$ where only $\gamma$, $\gamma_b$, or $\kappa_b$ have been updated to iteration $i + 1$ are noted $S_\gamma^i$, $S_{\gamma B}^i$, and $S_{\kappa B}^i$ respectively.

Starting with $\gamma$, we compute $S_\gamma^0$, and obtain $\gamma^2$ according to the following equation:

$$\gamma^{i+1} = \gamma^i - \alpha(S_\gamma^{i-1} - S^{i-1})/(\gamma^i - \gamma^{i-1}) \qquad (9)$$

Here $\alpha$ is a parameter that allows tuning how fast the algorithm can converge, and how precise the final result is. We typically use an initial value of $\alpha = 0.01$, but go down to $\alpha = 10^{-4}$ to fine-tune the final parameters. We follow the same steps for $\gamma_b^2$, $\kappa_b^2$, and finally compute $S^1$ for the next step. All subsequent parameters are then obtained by iterating on the previously described process. Eventually, the algorithm converges towards a final set of simulation parameters, that are then used in the simulations.

In Fig. 5a, we show a comparison between the experiments (thick faded lines) and the Langevin simulations (full lines), for $\langle W(t) \rangle$, $\langle E(t) \rangle$, and $\langle V(t) \rangle$ (blue, red, and orange respectively). All curves correspond to experiments with $t_{neq} = 1$ s, and simulations agree very well with the experiments. As an inset, we also show a typical example of the convergence process for the model parameters $\gamma^i$, $\gamma_b^i$, and $\kappa_b^i$, as a function of the iteration step (here $\alpha = 0.01$). All parameters used for the simulations are available in Table 3.

### Generalized Langevin equation
These two coupled Langevin equations, Eqs. (5) and (6), can be equivalently reformulated in terms of a single GLE:

$$\int_{-\infty}^t \Gamma(t - t')\dot{X}(t')dt' = -\nabla V + \nu(t) \qquad (10)$$

where the memory kernel is given by

$$\Gamma(t - t') = 2\gamma\delta(t - t') + \kappa_b e^{-(t-t')\kappa_b/\gamma_b}, \qquad (11)$$

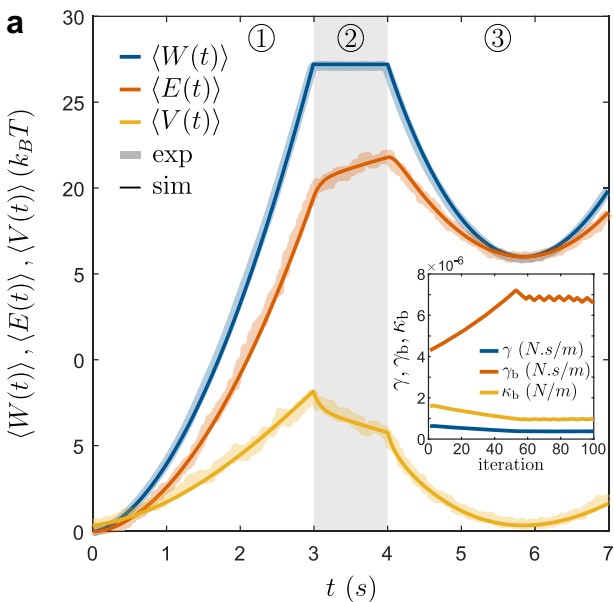

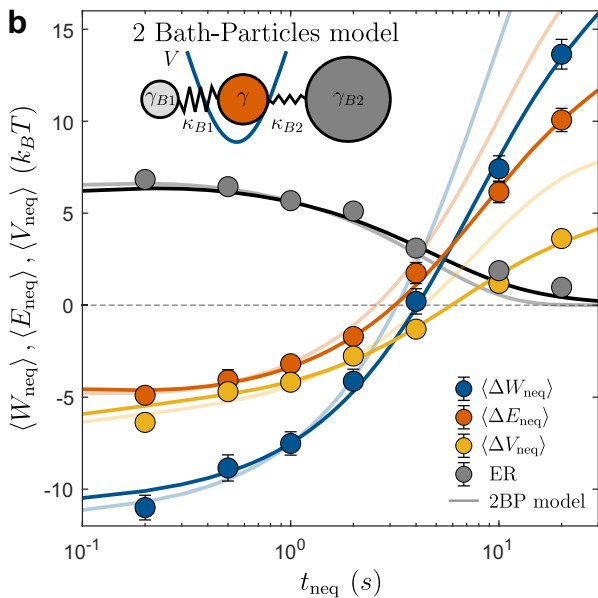

**Fig. 5 | Numerical simulations and 2 bath-particles model. a** $\langle W(t)\rangle$ (blue), $\langle E(t)\rangle$ (red), and $\langle V(t)\rangle$ (orange) for experiments (faded lines) and simulations (plain lines) after optimizing the parameters of the one bath-particle model. For a given value of $t_{\text{neq}}$, the model perfectly matches the experimental observations. Inset: evolution of the simulation parameters through the iterative process (only for the coarse initial process). **b** Comparison between the one and two bath-particles models for different values of $t_{\text{neq}}$. Symbols correspond to experiments, full lines to simulations with the two bath-particles model, and faded lines to the one bath-particle model. Compared to Fig. 2a, the extended model exhibits a close agreement with the experimental data across all values of $t_{\text{neq}}$, spanning two orders of magnitude.

### Table 3 | Parameters used for the two bath-particles model

| $\kappa$ ($\mu$N m$^{-1}$) | $\gamma$ ($\mu$N s m$^{-1}$) | $\gamma_b$ ($\mu$N s m$^{-1}$) | $\kappa_b$ ($\mu$N m$^{-1}$) | $\gamma_{b2}$ ($\mu$N s m$^{-1}$) | $\kappa_{b2}$ ($\mu$N m$^{-1}$) |
|---|---|---|---|---|---|
| 1.95 | 0.38 | 3.45 | 0.71 | 14.8 | 0.24 |

and the noise term $\nu(t)$ is a colored Gaussian noise with correlation

$$\langle \nu(t)\nu(t')\rangle = 2k_B T[\gamma\delta(t-t') + \gamma_b e^{-|t-t'|\kappa_b/\gamma_b}] \quad (12)$$

Additional bath particles (e.g., extensions of the micromechanical model, see below) provide further exponential terms to the memory kernel, thereby introducing multiple relaxation timescales. This hierarchical structure of relaxation is discussed, for instance, in Appendix E of ref. 23.

The memory kernel $\Gamma(t-t')$ can also be directly related to the viscoelastic shear modulus $G(t-t')$ of the fluid, through the generalized Stokes relation:

$$\Gamma(t-t') = 6\pi r G(t-t'), \quad (13)$$

where $r$ is the radius of the colloidal particle. In the frequency domain, this correspondence allows one to extract the storage and loss moduli, $G'(\omega)$ and $G''(\omega)$, commonly used in microrheology.

### Stochastic observables in the bath particle model
Starting from Eq. (5) and (6), we can directly apply the stochastic thermodynamics framework developed by Sekimoto[43] to derive key thermodynamic observables. The total internal energy of the system is given by the sum of the potential energy in the optical trap, $V$, and the elastic energy stored in the viscoelastic bath, $U_{\text{bath}}$:

$$U = V + U_{\text{bath}} = \frac{1}{2}\kappa(x-\lambda)^2 + \frac{1}{2}\kappa_b(x-x_b)^2 \quad (14)$$

From this, the stochastic work done on the system-arising from changes in the trap position $\lambda$-takes the standard form:

$$\delta W = \frac{\partial U}{\partial \lambda}d\lambda = -\kappa(x-\lambda)d\lambda \quad (15)$$

We can now explicitly compute the total dissipated heat $\delta Q$ as the difference between the work input and the change in internal energy:

$$\delta Q = \delta W - dV - dU_{\text{bath}} = -\frac{\partial V}{\partial x}dx - \frac{\partial U_{\text{bath}}}{\partial x}dx - \frac{\partial U_{\text{bath}}}{\partial x_b}dx_b. \quad (16)$$

Substituting the expressions for $V$ and $U_{\text{bath}}$, we obtain:

$$\delta Q = -\kappa(x-\lambda)dx - \kappa_b(x-x_b)dx + \kappa_b(x-x_b)dx_b. \quad (17)$$

This expression clearly separates into two components:

$$\delta Q_t = -(\kappa(x-\lambda) + \kappa_b(x-x_b))dx = (\gamma\dot{x} - \xi)dx \quad (18)$$

$$\delta Q_b = +\kappa_b(x-x_b)dx_b = (\gamma_b\dot{x}_b - \xi_b)dx_b \quad (19)$$

where $\delta Q_t$ accounts for dissipation due to the colloid's motion and $\delta Q_b$ captures dissipation from the relaxation of the bath degrees of freedom. These two terms reflect the partitioning of energy flow within the bath particle model.

### Two bath-particles model
In the present study, we opted for a single bath particle model to maintain simplicity in our approach. However, our recent work[38] demonstrated the necessity of incorporating a second relaxation time into the model to fully capture the relaxation process of our viscoelastic fluid. This straightforward extension involves adding a second bath particle, harmonically coupled to the colloidal particle (see Fig. 5b, inset), which introduces an additional relaxation time-scale. Consequently, we obtain a set of three coupled Langevin equations for

all three particles:

$$\gamma \dot{x}(t) = -\kappa_b(x - x_b) - \kappa_{b2}(x - x_{b2}) - \nabla V + \xi(t) \tag{20}$$

$$\gamma_b \dot{x}_b(t) = -\kappa_b(x_b - x) + \xi_b(t) \tag{21}$$

$$\gamma_{b2} \dot{x}_{b2}(t) = -\kappa_{b2}(x_{b2} - x) + \xi_{b2}(t) \tag{22}$$

The model parameters can be obtained from the experimental data using the same routine as described above. In Fig. 5b, we show the result of the model (full lines) and compare it to the experiments for different values of $t_{neq}$ (symbols). For an easier comparison, we show the previous results (see Fig. 2a) from the one bath-particle model as faded lines in the background. Clearly, the addition of a second time-scale allows to for a much better match of the experimental data, notably for large values of $t_{neq}$. As for the case with one bath-particle, these jointed equations can be combined in a single GLE, where the timescales simply adds together (see appendix of ref. 23). The set of parameters used with the two bath-particles model is available in Table 3.

## Data availability
The processed data used to generate the figures in this study are available on FigShare at the following link: https://doi.org/10.6084/m9.figshare.30365002. Due to the large size and complexity of the raw experimental data, they are only available from the corresponding author upon request.

## Code availability
The computer code developed for the numerical simulations presented in this work is available on FigShare at the following link: https://doi.org/10.6084/m9.figshare.30365002.

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

## Acknowledgements

We thank Christina Wege for providing the TMV solution. We thank Samuel Monter and Sarah Loos for fruitful discussions. The authors made limited use of Large Language Models (LLMs) to proofread the text and enhance the clarity and flow of the manuscript. This project was funded by the Deutsche Forschungsgemeinschaft (DFG), Grant No. SFB 1432—Project ID 425217212 (C.B.) and the ERC AdG BRONEB (101141477) (C.B.).

## Author contributions

F.G. performed the experiments, data analysis, and numerical simulations. C.B. contributed to the concept and design of the project. All authors contributed to the preparation of the manuscript.

## Funding

## Competing interests

The authors declare no competing interests.
