## [Transparent Peer Review file · Nature Communications]

Energy Recuperation of Driven Colloids in Non-Markovian Baths

Corresponding Author: Dr Felix Ginot

Version 0:

Reviewer comments:

Reviewer #1

(Remarks to the Author)

This work addresses an interesting problem concerning the way energy is dissipated and stored when moving a colloidal particle in a viscous or viscoelastic fluid. The paper, and in particular the introduction, is well-written and draws interest to the content. The experiments are carried out expertly, and the group has an excellent track record in the field. However, after reading the paper carefully, the authors were not able to convince me of the relevance of their main findings. The authors also omitted to put their work in the context of related fields, such as the broad and well-established fields of linear rheology of viscoelastic media and the extensive body of work on active microrheology that addresses similar topics. The authors claim that in a viscoelastic medium, in contrast to a purely viscous medium, energy is stored when displacing a particle actively by applying an optical tweezer force, provided the measurements are done on timescales shorter than the terminal relaxation time of the viscoelastic fluid. This finding looks pretty trivial, in my view. Isn't the energy stored in an elastic medium the same as the energy stored in the trap? (note that the trap potential corresponds to a Kelvin-Voigt viscoelastic solid material). The authors define many different functions to separate the storage and dissipation in the elastic fluid from the elastic trap, but is this really necessary to understand the main mechanism? I realize that there is always loss and storage in the viscoelastic medium (as the name says), and it is a bit complicated to describe the contribution of all the terms, but again, I feel like these things have been discussed and dealt with in the context of active microrheology for the reciprocal problem of extracting the viscoelastic functions $G'(w)$ and $G''(w)$ from the displacement curves. For a constant force, linear response theory shows that $\langle x(t)^2 \rangle = \frac{1}{kT} \times F$. I don't have the time to study this problem in detail, but I think the textbook by Furst and Squires on "Microrheology" could be a starting point, as well as the work by Puertas and Fuchs (eg PRL 102 (24), p.248302, 2009 and subsequent studies). Finally, the authors fail to convince me that the effect is useful for colloidal applications for the same reasons—it is simply the problem of a harmonically bound Brownian particle subjected to a time-dependent force. The terminal relaxation is irrelevant at the timescale considered here. Perhaps I missed the point, but the authors should address these concerns in simple terms; otherwise, I cannot recommend the publication of this paper.

Reviewer #2

(Remarks to the Author)

The authors of Energy Recuperation of Driven Colloids in Non-Equilibrium Baths have found a new way to measure elastic and viscous properties of surface induced order of complex fluids that in thermodynamic equilibrium evolves in the vicinity of a colloidal particle. The manuscript shows intriguing experiments, however this referee sees five major weaknesses in the theoretical interpretation as follows:[^]

1. the ignorance of theoretical work on surface induced order

It is long known that the structure of any liquid, complex or not is altered by the presence of a solid surface. In mixtures of two fluids a depletion layer depleted by one of the components forms around the solid colloid; in block copolymer systems the order of the blockcopolymer at the surface differs from that of the bulk; in liquid

crystalline free standing films one observes the phenomenon of surface induced freezing at the liquid crystalline/air-interface. In any case the presence of a surface reduces the entropy density of the complex fluid near the surface and is the driving force of surface induced order (SIO). It is thus natural to assume also a surface induced ordered region around the colloid embedded into the complex fluid.

The authors should mend this weakness with spending a prominent part of the introduction on surface induced order and cite the relevant literature.

2. the use of the term non-equilibrium bath

A bath or reservoir in thermodynamics is a thermodynamic equilibrium system infinitely larger than the thermodynamic system under investigation. It cannot be altered by the system because of its sheer size. Any sub-volume of a bath is equivalent to any other sub-volume somewhere else inside the bath. If you dump energy into the bath you cannot recover it because you did not change the bath by dumping the energy into it. The system has boundaries toward one or several baths. Each bath/system boundary imposes boundary conditions on the system. One usually does not know any details more than the equilibrium parameters of the bath. The focus of investigation is the system not the bath. So what the authors have shown that the bath is not the entire complex fluid around the colloidal particle. Some of the complex fluid around the colloidal particle is part of the system and not the bath and the recovery of the energy of the colloidal particle is due to energy flux from one part of the system to the other.

The authors should increase the system size of their model to include the surface induced ordered phase and abstain from the use of a non-equilibrium bath.

3. the use of a fictitious particle placed in the bath

The additional particle is neither fictitious (it has been measured by the authors) nor is it placed inside the bath (energy can be recovered from it, see point 1). To this referee the particle exists and it is placed into the system not the bath.

The authors should not use the attribute fictitious and they should not place the particle into the bath but into the system.

4. The focus on the energy recovery

Now that we know, that the particle is part of the system there is nothing wondrous about the exchange of elastic energy between the bare colloidal particle and the other particle to be identified in thermodynamic equilibrium as the SIO-dress to the bare colloidal particle. The bare colloidal particle with its SIO-dress constitute the dressed colloidal particle. What the authors have shown is that in complex fluids with long relaxation times they are able to partially undress the dressed bare colloidal particle without the dress immediately decaying and reforming at the displaced location. This allows them to probe the equilibrium and non equilibrium properties of the surface induced ordered phase using their optical manipulation technique.

Instead of wondering about the energy recovery the authors should focus on the SIO-properties they measure with their tweezers. The surface induced ordered phase probably would have an interfacial tension toward the solid and toward the bulk complex fluid, it would also properly have a interfacial viscosity at the dress/bulk boundary. Some of the fitting parameters of their micro-mechanical model have exactly those units.

5. the title of the paper

With the energy recovery demystified and the non-equilibrium bath abolished a new title is in order. May be viscoelastic equilibrium and non-equilibrium properties of a surface induced ordered region around a colloidal particle in a complex fluid or the like.

After lamenting about weaknesses here comes the praise: With their experiments the authors have for the first time succeeded in measuring properties of the surface induced ordered phase around a colloidal particle. These are breakthrough experiments. It is

for this reason this referee supports publication of this work after both authors have thought about my criticism. They can turn their work into a strong paper. Most of the changes can be done easily by renaming, i.e. using the proper names. The referee congratulates the authors for their nice findings.

Reviewer #4

(Remarks to the Author)

The paper by Ginot and Bechinger describes measurements of the “recuperation” of energy during the reverse motion of a microparticle in a viscoelastic fluid. The particle is grabbed and moved with an optical tweezer, and from the displacement out of the trap center, the force and stored potential energy are computed, and from the force integrated over the moved distance, the work performed.

1. While the measurements seem to be done with care, and the applied formalism looks ok, I find this paper quite confusing and misleading. If I understand the paper correctly, the bead is moved in a cycle, i.e. forth and back (with some waiting time at the reversal point, but then this is nothing else than an active microrheology experiment. And we know for an active microrheology experiment in a viscoelastic fluid (same as for a regular rheology experiment) that surely some of the performed energy is stored, and some is lost. In microrheology, to evaluate the performed work and lost energy, the force is typically plotted as a function of the distance moved (or in a regular rheology experiment, the stress as a function of strain) to produce Lissajous figures, from which the performed and dissipated energy can be determined as enclosed areas. While regular (micro)rheology experiments are typically performed under sinusoidal strain, and not in a triangular protocol as applied here, this does not change much. Also, the delay time is something that is sometimes done in (micro)rheology, known as stress relaxation: in a Lissajous diagram, it will lead to a lowering of the force/stress at the reversal point, so that on the way back, the system goes along a deeper-lying route, causing the enclosed area to be larger. It is not clear to me why the authors have not chosen this much more familiar and rich formalism to present their data. This literature is vast, and many examples have been studied, from biological tissue to artificial complex fluids and gels, and not only probe-time-dependent elastic/viscous moduli have been determined, but also the nonlinear response has been characterized, among others by the shape of the Lissajous curves, which is very insightful to elucidate the viscoelasticity and energy storage behavior of the “bath”.

2. In fact, as an example, the paper by Vitali et al., Sci. Rep. 10, 5831 (2020), studies, by active microrheology, the very same viscoelastic solution (Cetylpyridinium Chloride (CPyCl) and Sodium Salicylate (NaSal)) as the current article, and plots Lissajous curves that very clearly show energy storage on fast timescales, but gradually more energy loss (enlarged area of the Lissajous curve) as the probe time scale (parametrized as the probing frequency) increases.

3. In this sense, I also find the introduction misleading; it is extremely short and doesn't even mention the rich field of microrheology with its energy storage and loss. As this is a more than 20 year-old field that seems to have crucial relevance and strong overlap with the presented study, it should be clearly framed in the introduction, and distinction/motivation of the current work clearly discussed in view of this context. But maybe I'm misunderstanding or missing something.

4. The modelling can then probably also be done in a usual way with a regular viscoelastic spring-dashpot model. Probably the presented model is equivalent to that, but then again, why is the usual viscoelastic formalism not applied? The paper seems to use different semantics for the same physics that has been formulated many times.

5. I also find the analogy with the moving piston misleading: while of course there is some friction in the piston, which somewhat lowers the efficiency of a heat engine, the limit of its efficiency has fundamental entropic reasons that are very different from additional friction effects. As the authors define an efficiency measure in the next sentence, this context is quite confusing. The present article does not study an engine that runs on its own, but simply a forth- and back motion performed by external means. In this sense, Sigma should probably not be called efficiency (which is typically a number between 0 and 100%), but something like energy storage capacity.

Due to the above points, I cannot recommend the paper for publication in its current state, and, written in more familiar terms, it may no longer be suitable for Nat. Comm.

Version 1:

Reviewer comments:

Reviewer #1

(Remarks to the Author)

Reviewer #2

(Remarks to the Author)

The authors agree with the reviewer that one cannot measure whether a degree of freedom is a degree of freedom of the bath or the system.

The declaration of a particle being part of the bath is a choice made by the author. Two coupled oscillators periodically exchange energy as a function of time. Declaring one of the oscillators as part of the system the other as part of the bath causes energy to be recuperated periodically from the bath (in their work the two oscillators are also coupled to a thermodynamic reservoir to make the recuperation less obvious). No measurement needs to be done to discover such energy recuperation. The fact that the authors recuperate energy from a bath should not be surprising, considering their choice of the bath: a viscoelastic fluid parts of which are elastic and exchange energy with the system.

The energy recuperation would be surprising and even in violation with thermodynamic principles if the choice of bath would be a thermodynamic reservoir, where by definition it is impossible to recuperate energy. The authors, however, depict the energy recuperation as something astonishing which it is clearly not. The authors do not need any of their experiments to get their energy recuperation. They merely must define their bath appropriately.

All three referees complained about the misleading language used by the authors. I still think the experiments are interesting measurements of the surface induced order of the viscoelastic fluid around the particle. The most disqualifying statement about the work comes from the authors themselves:
I quote:

Because U_{bath} is stored in the hidden degrees of freedom of the bath, it is by definition not accessible experimentally.

The authors give up on trying to measure the cause of their observation by defining it to be experimentally not accessible. This is a complete capitulation of an experimentally working group.
I am sure it is not easy to access the microscopic properties of the bath, but I am sure where there is a will there is a way.

In summary the authors are running into a dead end focussing on energy recuperation from a so defined non markovian bath. It is foremost in the interest of the authors that this work should not be published since it would ruin their reputation. I think the work should be rejected in its current form.

Reviewer #5

(Remarks to the Author)
See the attached file.

Version 2:

Reviewer comments:

Reviewer #3

(Remarks to the Author)
The authors have responded convincingly to all my questions and addressed all my concerns. Therefore, I recommend publication of the manuscript in its present form in Nature Communications.

Reviewer #5

(Remarks to the Author)
The authors have satisfactorily addressed all my comments, and I am happy to recommend the manuscript for publication.

Answer to the Reviewers

Reviewer #1 (Remarks to the Author):

This work addresses an interesting problem concerning the way energy is dissipated and stored when moving a colloidal particle in a viscous or viscoelastic fluid. The paper, and in particular the introduction, is well-written and draws interest to the content. The experiments are carried out expertly, and the group has an excellent track record in the field.

We thank the reviewer for appreciating the quality of the writing and experiments.

However, after reading the paper carefully, the authors were not able to convince me of the relevance of their main findings. The authors also omitted to put their work in the context of related fields, such as the broad and well-established fields of linear rheology of viscoelastic media and the extensive body of work on active microrheology that addresses similar topics.

Similar to many studies, we used a colloidal probe confined in a complex fluid by a harmonic potential. In this regard we fully understand why the referee could expect our work to be put in the context of (micro)rheology.

However, our approach differs fundamentally: rather than using the particle to probe the fluid's properties, we investigate how a non-Markovian bath modifies the energetics of the particle in the trap. In this context, the viscoelastic fluid only acts as a well-known example of a bath with memory effects. Our focus is then on the consequences of the time-delayed response compared to the commonly studied Newtonian case, not on characterizing its microscopic origins. To confirm the generality of our findings, we also examined two additional fluids (a PMMA solution and a TMV suspension), both of which exhibited qualitatively similar behavior.

From this perspective, we believe that Stochastic Thermodynamics provides the most appropriate framework for analyzing energy exchanges in our system. We acknowledge that this distinction was not sufficiently clear in our original submission. To address this, we have revised key sections of the manuscript to explicitly emphasize the non-Markovian nature of the bath while de-emphasizing its viscoelastic properties. In particular we have rewritten the introduction completely, acknowledging the well-known storage and loss modulus of viscoelastic fluids.

Changes: *Introduction was fully rewritten.*

The authors claim that in a viscoelastic medium, in contrast to a purely viscous medium, energy is stored when displacing a particle actively by applying an optical tweezer force, provided the measurements are done on timescales shorter than the terminal relaxation time of the viscoelastic fluid. This finding looks pretty trivial, in my view. Isn't the energy stored in an elastic medium the same as the energy stored in the trap? (note that the trap potential corresponds to a Kelvin-Voigt viscoelastic solid material).

We fully agree with the reviewer that displacing a particle in a viscoelastic fluid results in both energy loss and storage, a well-known phenomenon that we have already addressed in the introduction. As a side note, the energy stored in the bath equals that within the trap only at steady state, which is generally not the case in our study. Moreover, the mere presence of

stored energy in the bath does not necessarily imply that it can be extracted as meaningful work.

However, our statement is more general. First, in a non-Markovian, slow-relaxing bath, excitations remain temporarily localized and can be reabsorbed by the particle before dissipating. Second, this storage-recovery behavior is maximized when the relaxation time of the bath matches the system's intrinsic timescale—an effect with significant implications for operating in such environments. Again, we used a viscoelastic bath only as a representative example to illustrate this phenomenon. To clarify these points, we have revised the manuscript accordingly, and in particular fully rewritten the Discussion part.

Changes: *Discussion was fully rewritten*

The authors define many different functions to separate the storage and dissipation in the elastic fluid from the elastic trap, but is this really necessary to understand the main mechanism? I realize that there is always loss and storage in the viscoelastic medium (as the name says), and it is a bit complicated to describe the contribution of all the terms, but again, I feel like these things have been discussed and dealt with in the context of active microrheology for the reciprocal problem of extracting the viscoelastic functions $G'(w)$ and $G''(w)$ from the displacement curves. For a constant force, linear response theory shows that $\langle x(t)^2 \rangle = \langle x^2(t) \rangle_{eq} / kT \times F$. I don't have the time to study this problem in detail, but I think the textbook by Furst and Squires on "Microrheology" could be a starting point, as well as the work by Puertas and Fuchs (eg PRL 102 (24), p.248302, 2009 and subsequent studies).

In microrheology, it is common to use G' and G'' to characterize the viscoelastic properties of a system, as the referee points out. However, this approach is most relevant when the system is subjected to oscillatory shear and has reached a steady state—conditions that do not apply to our study.

In this context, the use of a generalized Langevin equation with a decaying memory kernel is more appropriate. This approach is strictly equivalent to our tracer-bath particle model, as mentioned on p.6 and refs [26, 27]. Moreover, as shown in previous work (see Gómez-Solano & Bechinger, *New Journal of Physics*, 2015), deriving the memory kernel from measurements of G' and G'' obtained via conventional active microrheology, or directly fitting transient and equilibrium measurements, yields consistent results.

Ultimately, this is a matter of choosing the right tools for the right problem. Since our study focuses on the energetics of a particle driven by a trap within a complex bath, we find it more appropriate to use observables such as friction and relaxation time rather than G' and G'' , particularly because we aim to compute stochastic thermodynamic observables. We added a discussion regarding our choice of using a tracer-bath particle model, instead of the G' G'' functions typically used in microrheology.

p4: *"Note that these two coupled Markovian equations can be combined into a single generalized Langevin equation [45, 46] with an exponentially decaying memory kernel (for an explicit derivation, see [22]). From this memory kernel, one can derive the storage modulus G' and the loss modulus G'' [47], which are commonly used in microrheology to characterize viscoelastic fluids. In contrast, the present micro-mechanical model provides a more intuitive picture of the hidden degrees of freedom of the bath and enables straightforward derivations of the different stochastic observables, thereby justifying its use in our study."*

Finally, the authors fail to convince me that the effect is useful for colloidal applications for the same reasons—it is simply the problem of a harmonically bound Brownian particle subjected to a time-dependent force. The terminal relaxation is irrelevant at the timescale considered here. Perhaps I missed the point, but the authors should address these concerns in simple terms; otherwise, I cannot recommend the publication of this paper.

Because energy can not only be stored in the bath but also reinjected into the particle, this leads to a significant reduction in the friction force experienced by the particle. Importantly, we have demonstrated that this effect can be maximized by tuning the protocol timescale to match that of the fluid, or conversely, by selecting a fluid with the appropriate relaxation time to optimize a given protocol.

We also want to stress that the force acting on the particle is not merely a time-dependent external drive but rather a complex interplay between the optical trapping force, particle friction, and the delayed response of the bath.

Lastly, regarding the presence of a terminal relaxation is necessary to investigate and discuss the dissipation of the excitation. In the experiments, t_{neq} always has a minimum value due to the finite response time of the piezo-driven stage, which is constrained by inertia. As a result, it is impossible to instantaneously reverse the motion of the trap.

p2: *“Note that experimentally there is a minimum value of $t_{\text{neq}} > 0.1\text{s}$ due to the finite response time of the stage.”*

We thank the referee for their remarks, which helped us recognize that our key message was not as clear as we intended. We believe the revised manuscript now conveys the originality of our study more effectively. In particular, we have made a greater effort to explicitly justify how our work differs from a classical study of the fluid's loss and storage moduli. We hope that these substantial revisions address the referee's concerns and lead to a reassessment of their judgment.

Reviewer #2 (Remarks to the Author):

The authors of Energy Recuperation of Driven Colloids in Non-Equilibrium Baths have found a new way to measure elastic and viscous properties of surface induced order of complex fluids that in thermodynamic equilibrium evolves in the vicinity of a colloidal particle. The manuscript shows intriguing experiments, however this referee sees five major weaknesses in the theoretical interpretation as follows:

1. the ignorance of theoretical work on surface induced order

It is long known that the structure of any liquid, complex or not is altered by the presence of a solid surface. In mixtures of two fluids a depletion layer depleted by one of the components forms around the solid colloid; in block copolymer systems the order of the block copolymer at the surface differs from that of the bulk; in liquid crystalline free standing films one observes the phenomenon of surface induced freezing at the liquid crystalline/air-interface. In any case the presence of a surface reduces the entropy density of the complex fluid near the surface and is the driving force of surface induced order (SIO). It is thus natural to assume also a surface induced ordered region around the colloid embedded into the complex fluid.

The authors should mend this weakness with spending a prominent part of the introduction on surface induced order and cite the relevant literature.

We fully agree with the reviewer that the mere presence of a particle in a fluid—even at equilibrium—leads to a local coupling which presents different properties from the bulk. Such effects can lead to the rise of memory and thus non-Markovian dynamics. In fact, in ref [15], we identified the presence of a second timescale in the same fluid—one that is typically absent at the macroscale. There, we briefly discussed its possible origin, suggesting that it may arise from local coupling between the colloid's surface and the bulk micellar network. In the present study, where displacements are small, we could imagine this effect to be dominant, which could explain why a single timescale provides a sufficiently accurate description of the system.

However, we emphasize that the precise origin of the bath's time-delayed response is not crucial here. What matters is that the bath exhibits sufficiently strong and slow non-Markovian properties, allowing them to be studied in detail.

We have rewritten the introduction, mentioning that non-Markovian behavior can originate from the complex coupling between the particle and the bath, and citing SIO as an example.

Changes: *Introduction was fully rewritten.*

2. the use of the term non-equilibrium bath

A bath or reservoir in thermodynamics is a thermodynamic equilibrium system infinitely larger than the thermodynamic system under investigation. It cannot be altered by the system because of its shear size. Any sub-volume of a bath is equivalent to any other sub-volume somewhere else inside the bath. If you dump energy into the bath you cannot recover it because you did not change the bath by dumping the energy into it. The system has boundaries toward one or several baths. Each bath/system boundary imposes boundary conditions on the system. One usually does not know any details more than the equilibrium

parameters of the bath. The focus of investigation is the system not the bath. So what the authors have shown that the bath is not the entire complex fluid around the colloidal particle. Some of the complex fluid around the colloidal particle is part of the system and not the bath and the recovery of the energy of the colloidal particle is due to energy flux from one part of the system to the other.

A bath is typically defined as a reservoir with infinite heat capacity that remains in thermodynamic equilibrium. However, this does not preclude the emergence of local deviations from equilibrium, which commonly occur when the bath's response is not instantaneous—a feature inherent to any real-world system. The concept of a non-Markovian bath, characterized by a slowly decaying memory kernel, helps bridge this gap and has been widely employed in both theoretical (e.g., Speck & Seifert, *J. Stat. Mech.*, 2007) and experimental studies (e.g., Mehl et al., *Phys. Rev. Lett.*, 2012).

The authors should increase the system size of their model to include the surface induced ordered phase and abstain from the use of a non-equilibrium bath.

3. the use of a fictitious particle placed in the bath

The additional particle is neither fictitious (it has been measured by the authors) nor is it placed inside the bath (energy can be recovered from it, see point 1). To this referee the particle exists and it is placed into the system not the bath.

The authors should not use the attribute fictitious and they should not place the particle into the bath but into the system.

We refer to our bath particle as *fictitious* in contrast to the *tracer particle*, which represents the colloid itself—a formalism we have applied in previous works (see refs [15, 24]. Discussing the system's boundaries is indeed an interesting perspective.

As the reviewer suggests, one could redefine the system to include both the particle and all hidden degrees of freedom responsible for the bath's non-Markovian properties. In this idealized framework, all observables would be perfectly known, the environment would act as a true equilibrium bath, and classical Markovian dynamics would be recovered. However, such a complete microscopic description is only feasible in theory and typically fails when applied to complex experimental systems.

In this context, we believe that our approach—coarse-graining the slow-decaying hidden degrees of freedom into a non-Markovian bath—offers particularly practical and valuable insights.

We added a short paragraph discussing the notion of non-Markovian bath, and our choice to coarse-grain the slow-decaying hidden degrees of freedom of the environment inside the bath itself.

p4: “As in the experiments, the system under study is limited to the colloidal particle and the harmonic potential. Alternatively, one could include the bath particle within the system itself. In this idealized framework, all observables would be fully determined, the environment would behave as a true equilibrium bath, and classical Markovian dynamics would be recovered. However, such a complete microscopic description requires direct access to the

hidden degrees of freedom, which is theoretically possible but often impractical for complex experimental systems. Given these limitations, we argue that our coarse-graining approach—integrating the slow-decaying hidden degrees of freedom into a non-Markovian bath—provides a more practical and insightful framework.”

4. The focus on the energy recovery

Now that we know, that the particle is part of the system there is nothing wondrous about the exchange of elastic energy between the bare colloidal particle and the other particle to be identified in thermodynamic equilibrium as the SIO-dress to the bare colloidal particle. The bare colloidal particle with its SIO-dress constitute the dressed colloidal particle. What the authors have shown is that in complex fluids with long relaxation times they are able to partially undress the dressed bare colloidal particle without the dress immediately decaying and reforming at the displaced location. This allows them to probe the equilibrium and non equilibrium properties of the surface induced ordered phase using their optical manipulation technique.

Instead of wondering about the energy recovery the authors should focus on the SIO-properties they measure with their tweezers. The surface induced ordered phase probably would have an interfacial tension toward the solid and toward the bulk complex fluid, it would also properly have a interfacial viscosity at the dress/bulk boundary. Some of the fitting parameters of their micro-mechanical model have exactly those units.

We are not entirely certain that we fully understand the referee's point. However, we agree that energy exchange between the particle and the viscoelastic bath, in itself, is not surprising. The key focus of our study is on **how** and **how much** energy can be stored and recovered from the bath.

Regarding the SIO effect mentioned by the referee, we do not believe that our current study is the most suitable framework to highlight these effects, as we primarily use a single bath particle model in the main text. From our understanding, such an effect would introduce an additional timescale beyond the intrinsic relaxation time of the micellar network. In this context, we encourage the referee to refer to our previous works ([refs 15, 25]), where we extensively discuss the emergence of this additional timescale and demonstrate how the presence or absence of a trap leads to distinct sets of reciprocal and non-reciprocal timescales.

5. the title of the paper

With the energy recovery demystified and the non-equilibrium bath abolished a new title is in order. May be viscoelastic equilibrium and non-equilibrium properties of a surface induced ordered region around a colloidal particle in a complex fluid or the like.

After reviewing the referees' reports and noting their emphasis on the viscoelastic aspects of our study, we have decided to revise the title to better reflect our main focus: **"Energy Recuperation of Driven Colloids in non-Markovian Baths."**

After lamenting about weaknesses here comes the praise: With their experiments the authors have for the first time succeeded in measuring properties of the surface induced ordered phase around a colloidal particle. These are breakthrough experiments. It is for this reason this referee supports publication of this work after both authors have thought about my criticism. They can turn their work into a strong paper. Most of the changes can be done easily by renaming , i.e. using the proper names. The referee congratulates the authors for their nice findings.

We appreciate the referee's kind remarks, although we are unsure they are entirely warranted. We also thank the referee for highlighting the SIO properties, which could indeed open new avenues for investigating non-Markovian environments in the future. However, we trust the referee will understand our decision not to focus on this specific effect, as the precise origin of the system's memory is not central to the objectives of this study. We believe the substantial revisions we made to the manuscript now clarify the main point of our work.

Reviewer #4 (Remarks to the Author):

The paper by Ginot and Bechinger describes measurements of the “recuperation” of energy during the reverse motion of a microparticle in a viscoelastic fluid. The particle is grabbed and moved with an optical tweezer, and from the displacement out of the trap center, the force and stored potential energy are computed, and from the force integrated over the moved distance, the work performed.

1. While the measurements seem to be done with care, and the applied formalism looks ok, I find this paper quite confusing and misleading. If I understand the paper correctly, the bead is moved in a cycle, i.e. forth and back (with some waiting time at the reversal point, but then this is nothing else than an active microrheology experiment. And we know for an active microrheology experiment in a viscoelastic fluid (same as for a regular rheology experiment) that surely some of the performed energy is stored, and some is lost. In microrheology, to evaluate the performed work and lost energy, the force is typically plotted as a function of the distance moved (or in a regular rheology experiment, the stress as a function of strain) to produce Lissajous figures, from which the performed and dissipated energy can be determined as enclosed areas. While regular (micro)rheology experiments are typically performed under sinusoidal strain, and not in a triangular protocol as applied here, this does not change much. Also, the delay time is something that is sometimes done in (micro)rheology, known as stress relaxation: in a Lissajous diagram, it will lead to a lowering of the force/stress at the reversal point, so that on the way back, the system goes along a deeper-lying route, causing the enclosed area to be larger. It is not clear to me why the authors have not chosen this much more familiar and rich formalism to present their data. This literature is vast, and many examples have been studied, from biological tissue to artificial complex fluids and gels, and not only probe-time-dependent elastic/viscous moduli have been determined, but also the nonlinear response has been characterized, among others by the shape of the Lissajous curves, which is very insightful to elucidate the viscoelasticity and energy storage behavior of the “bath”.

The reviewer is correct that our system operates on a cycle with waiting times. In this sense, it can indeed be compared to a (micro)rheology experiment, a field with a well-established and extensive literature. However, it is important to note that Lissajous curves are typically employed when the system is in a (non-equilibrium) steady state, which is not the case here (the system starts at and relaxes to equilibrium at the beginning and the end of the cycle).

More generally, as mentioned above, we deliberately minimized references to microrheology because our focus is not on characterizing the material properties of a specific viscoelastic fluid. Instead, we investigate the energetics of a well-known system—a particle in a harmonic potential—driven through a cycle within a non-Markovian bath. To emphasize the generality of our findings, we examined three distinct fluids: micelles (in the main manuscript), as well as a polymer solution and a TMV suspension (in the Supplementary Information), all of which exhibit the same qualitative behavior.

As the referee pointed out, when a particle moves within a viscoelastic bath, some energy is naturally stored while some is dissipated. We do not claim this is a novel observation, which is why energy storage was already discussed in the introduction. However, the key question is not just whether energy is stored but how and how much can be recovered and converted into useful work. In this context, the complex and non-monotonic dependence of energy recovery on the cycle timescale, as shown in Fig. 3, was particularly unexpected and, in our

view, a central finding of this study.

To clarify these points, we have revised the manuscript accordingly.

p6: *“Naturally, the fact that when driving a particle in a viscoelastic fluid some of the work is temporarily stored in the bath is fully expected. However, here we have demonstrated that this energy can then be reinjected to the particle, to perform useful work. Energy Recuperation can thus help to reduce frictional losses, which are typically encountered by micromachines operating in fluid environments.”*

2. In fact, as an example, the paper by Vitali et al., Sci. Rep. 10, 5831 (2020), studies, by active microrheology, the very same viscoelastic solution (Cetylpyridinium Chloride (CPyCl) and Sodium Salicylate (NaSal)) as the current article, and plots Lissajous curves that very clearly show energy storage on fast timescales, but gradually more energy loss (enlarged area of the Lissajous curve) as the probe time scale (parametrized as the probing frequency) increases.

We sincerely thank the reviewer for bringing this study to our attention. However, we note that the micelle concentration used in that study is an order of magnitude higher than in our case, making even a qualitative comparison challenging.

More importantly, as the reviewer pointed out, energy storage in that study decreases monotonically with increasing time scales. This appears to contradict our findings, where energy recovery (and storage) exhibits a non-monotonic behavior. This apparent paradox can be resolved by considering the presence of the trap, which provides an additional pathway for energy storage.

In summary, our study focuses on the energetics of a particle inside a harmonic potential, driven within a viscoelastic (non-Markovian) bath—not solely on the properties of the environment.

p6: *“Naturally, when a particle is driven in a viscoelastic fluid, some of the work is temporarily stored in the bath—a well-known consequence of the fluid’s storage modulus, often illustrated using Lissajous curves [48]. However, we have shown that this stored energy can subsequently be reinjected into the particle, allowing it to perform useful work.”*

3. In this sense, I also find the introduction misleading; it is extremely short and doesn’t even mention the rich field of microrheology with its energy storage and loss. As this is a more than 20 year-old field that seems to have crucial relevance and strong overlap with the presented study, it should be clearly framed in the introduction, and distinction/motivation of the current work clearly discussed in view of this context. But maybe I’m misunderstanding or missing something.

We aimed to make our introduction as clear and concise as possible, which is why it is relatively brief. However, we acknowledge that the balance between precision and completeness may not have been optimal, as evidenced by the referee’s concerns. In light of this, we agree that incorporating additional context from the well-established field of (micro)rheology would be beneficial. Providing a clearer distinction between our work and traditional microrheology will help better articulate our motivation and place our study within the broader scientific landscape.

We have therefore revised the introduction to address this point.

Changes: *Introduction was fully rewritten.*

4. The modelling can then probably also be done in a usual way with a regular viscoelastic spring-dashpot model. Probably the presented model is equivalent to that, but then again, why is the usual viscoelastic formalism not applied? The paper seems to use different semantics for the same physics that has been formulated many times.

The tracer-bath particle formalism we used is indeed perfectly equivalent to a spring-dashpot model, ie: a memory kernel with exponentially decaying memory (see p.6). We believe the tracer-bath particle formalism (which we have used in previous studies as in [15, 24]) offers a better and more intuitive insight regarding the dynamics of the system, and a clear separation between the Markovian and non-Markovian processes.

We added a short comment explaining why we opted for a tracer-bath particle description instead of using a generalized Langevin equation.

p4: *“Note that these two coupled Markovian equations can be combined into a single generalized Langevin equation [45, 46] with an exponentially decaying memory kernel (for an explicit derivation, see [22]). From this memory kernel, one can derive the storage modulus G' and the loss modulus G'' [47], which are commonly used in microrheology to characterize viscoelastic fluids. In contrast, the present micro-mechanical model provides a more intuitive picture of the hidden degrees of freedom of the bath and enables straightforward derivations of the different stochastic observables, thereby justifying its use in our study.”*

5. I also find the analogy with the moving piston misleading: while of course there is some friction in the piston, which somewhat lowers the efficiency of a heat engine, the limit of its efficiency has fundamental entropic reasons that are very different from additional friction effects. As the authors define an efficiency measure in the next sentence, this context is quite confusing. The present article does not study an engine that runs on its own, but simply a forth- and back motion performed by external means. In this sense, Sigma should probably not be called efficiency (which is typically a number between 0 and 100%), but something like energy storage capacity.

We fully agree with the referee that the efficiency of a heat engine is fundamentally constrained by entropy production, which is distinct from the additional friction discussed in our manuscript. Our analogy aimed to illustrate how non-ideal engines invariably experience extra dissipation—particularly due to friction—and how energy recuperation could be beneficial in this context.

Following the referee’s suggestion, we have revised this paragraph for clarity and renamed "efficiency" as "work ratio" to better reflect the intended concept.

p6: *“Naturally, the fact that when driving a particle in a viscoelastic fluid some of the work is temporarily stored in the bath is fully expected. However, here we have demonstrated that this energy can then be reinjected to the particle, to perform useful work. Energy Recuperation can thus help to reduce frictional losses, which are typically encountered by micromachines operating in fluid environments.”*

Due to the above points, I cannot recommend the paper for publication in its current state, and, written in more familiar terms, it may no longer be suitable for Nat. Comm.

We hope that the substantial revisions to the manuscript, particularly in the introduction and discussion, effectively address the referee's concerns. In particular, the revised wording and emphasis on stochastic thermodynamics now better justify our choice not to use a microrheology formalism. We trust that these improvements clarify our approach and strengthen the manuscript.

Point by point answer to the reviewers

Reviewer #2

The authors agree with the reviewer that one cannot measure whether a degree of freedom is a degree of freedom of the bath or the system. The declaration of a particle being part of the bath is a choice made by the author. Two coupled oscillators periodically exchange energy as a function of time. Declaring one of the oscillators as part of the system the other as part of the bath causes energy to be recuperated periodically from the bath (in their work the two oscillators are also coupled to a thermodynamic reservoir to make the recuperation less obvious). No measurement needs to be done to discover such energy recuperation.

The fact that the authors recuperate energy from a bath should not be surprising, considering their choice of the bath: a viscoelastic fluid parts of which are elastic and exchange energy with the system.

We thank the reviewer for their comment and agree that, in principle, it is impossible to experimentally determine whether a given degree of freedom belongs to the system or the bath. The designation of certain degrees of freedom as part of the “bath” is indeed a modeling choice. As mentioned by the referee, in the simple case of two coupled oscillators, energy is periodically exchanged between them. Labeling one oscillator as the system and the other as the bath inevitably leads to energy flowing back from the bath to the system.

However, this is a rather theoretical perspective that overlooks a key experimental constraint: not all degrees of freedom are easily observable in measurements. Therefore, it is meaningful to distinguish between observable degrees of freedom (easily accessible in experiments) and hidden degrees of freedom (inaccessible or at least very difficult to measure experimentally). In our study, we defined the system—the colloidal particle—such that its dynamics are observable with high accuracy using optical techniques. In contrast, the bath contains hidden degrees of freedom (microscopic micellar structure) that cannot be directly measured easily but are inferred through appropriate modeling. To clarify this important distinction, we have revised the manuscript to emphasize the roles of observable and hidden degrees of freedom.

Regarding the reviewer’s point that energy recuperation in a viscoelastic system is unsurprising, we respectfully disagree. While a driven colloidal particle can indeed transfer energy to the bath by exciting hidden degrees of freedom, this alone does not guarantee that energy will be recuperated by the system. For example, in a simple recoil experiment where the particle is first optically driven through the viscoelastic fluid and then released from the optical trap, no energy is transferred back during the particle’s relaxation. This outcome reflects the definition of work in small systems, which ensures a consistent framework in stochastic thermodynamics. To achieve actual reinjection of energy from the bath, the system must be placed in a suitable external potential—precisely what we implemented in our experiments.

The energy recuperation would be surprising and even in violation with thermodynamic principles if the choice of bath would be a thermodynamic reservoir, where by definition it is

impossible to recuperate energy. The authors, however, depict the energy recuperation as something astonishing which it is clearly not. The authors do not need any of their experiments to get their energy recuperation. They merely must define their bath appropriately.

We agree that energy recuperation from a classical (equilibrated) thermodynamic reservoir would indeed violate the second law of thermodynamics. However, in our work the bath is not a conventional thermodynamic reservoir, i.e. an inert thermostat, but a viscoelastic medium with elastic components that can transiently store and release energy. This distinction is crucial: unlike an ideal reservoir, our bath possesses memory and internal degrees of freedom capable of reversible energy exchange with the system.

We respectfully disagree, however, with the assertion that our experimental findings are not surprising and could be anticipated purely from an appropriate definition of the bath. While it is true that the system–bath partition is a modeling choice (see our response above) , the recuperation of energy is not a trivial consequence of this choice. In many viscoelastic systems, as illustrated for example by a recoil experiment (where a driven particle is released from its trap), no energy is transferred back to the system during relaxation. This outcome reflects the established framework of stochastic thermodynamics, where work is carefully defined to avoid ambiguities in small systems (see e.g. U. Seifert, "Stochastic thermodynamics, fluctuation theorems and molecular machines." *Reports on progress in physics* 75.12 (2012): 126001.).

What makes our experiments significant is that they implement an external potential tailored to exploit the bath's viscoelastic properties, thereby enabling controlled energy reinjection. This is not a generic feature of viscoelastic media but a result of our specific experimental design. We have revised the manuscript to clarify this point and to avoid any suggestion that energy recuperation would occur in violation of thermodynamic principles.

All three referees complained about the misleading language used by the authors. I still think the experiments are interesting measurements of the surface induced order of the viscoelastic fluid around the particle.

None of the referees raised concerns about misleading language; instead, they requested further clarification and explanation in specific areas. We have revised the manuscript accordingly to address each of their comments in detail.

As a side note, we would like to clarify that the concept of *surface-induced order* has never mentioned by us and seems not central to our interpretation of the results. Rather, the observed energy recuperation arises from the local compression of the viscoelastic fluid, which can store mechanical energy due to its long relaxation time. This interpretation is fully supported by our micromechanical model, which does not incorporate any ordering effects.

The most disqualifying statement about the work comes from the authors themselves: I quote: Because U_{bath} is stored in the hidden degrees of freedom of the bath, it is by definition not accessible experimentally.

We are unsure why the referee considers this sentence as disqualifying. By definition, hidden degrees of freedom cannot be directly observed (see also our answer below).

The authors give up on trying to measure the cause of their observation by defining it to be experimentally not accessible. This is a complete capitulation of an experimentally working group. I am sure it is not easy to access the microscopic properties of the bath, but I am sure where there is a will there is a way.

We believe there is a fundamental misunderstanding regarding the meaning of hidden degrees of freedom, which—by definition—are not directly accessible in experiments. Far from capitulating to this limitation, our experiments demonstrate the existence of such hidden degrees of freedom in our system. Moreover, we show that these experimentally inaccessible components can be effectively accounted for through a theoretical model that achieves excellent agreement with our data.

In summary the authors are running into a dead end focussing on energy recuperation from a so defined non markovian bath. It is foremost in the interest of the authors that this work should not be published since it would ruin their reputation. I think the work should be rejected in its current form.

We thank the reviewer for sharing their perspective. While we respect their opinion, we strongly disagree with the assertion that our focus on energy recuperation from a non-Markovian bath represents a “dead end.” On the contrary, we believe our work highlights a fundamental and experimentally relevant aspect of system–bath interactions in viscoelastic media, which are inherently non-Markovian. Our view is also shared by the other referees who find our work interesting and of immediate relevance for applications.

Reviewer #3

The manuscript reports an elegant experimental study on driven colloidal dynamics in viscoelastic liquids, where the external driving is realised by using optical tweezers. For a judiciously designed 4-steps driving protocol, which combines alternating driving and halting stages of different durations, a partially reversible energy exchange between the work reservoir and the viscoelastic bath is being reported. The main mechanism of the energy recuperation is related to the wide separation of the relaxation time of the host fluid and that of the colloidal particle, with the latter being much shorter as compared to the former. This allows for a temporary energy storage in and recovery from the slow degrees of freedom describing the viscoelastic medium.

The authors demonstrate a universality of this energy storage and recovery effect by considering three types of viscoelastic fluids with quite different relaxation mechanisms and times. They also show the absence of the effect when a conventional molecular Newtonian fluid, such as water-glycerol mixture, is used as the colloidal host. Because the relaxation time of the mixture is several orders of magnitude shorter than the time scales in the driving protocol, the fluid may be considered at thermodynamic equilibrium at any instantaneous position of the driven colloidal particle, and all the injected work is immediately dissipated as heat in the bath. The authors could explain their experimental results semi-quantitatively by using a very simple effective Langevin model of the colloidal particle moving through a viscoelastic fluid. Considering the simplicity of the model, the agreement between the theory and the experiments is fascinating. The Langevin model, formulated in terms of a pair of particles connected by an elastic spring and being characterised by different relaxation times, provides a clear and simple picture of the basic physics underlying the reported energy recuperation effect.

On my opinion this study will be of interests for a broad readership of Nature Communications working in the areas of non-equilibrium statistical mechanics, active matter, molecular motors and micromechanics. The presented results points towards the ways in which energy losses of driven microparticles can be reduced and the useful work of colloidal “motors” enhanced. Therefore, beyond pure academic interest, this study may have an immediate impact on the development of novel microfluidic application utilising driven or self-propelled colloidal particles as microengines. Based on this, I would recommend the publication of this manuscript in Nature Communications, provided the authors have addressed several technical comments and questions listed below.

We thank the reviewer for their positive assessment of our study and recommending our work for publication in Nature Communications after addressing remaining issues. We are particularly pleased that the reviewer highlighted the elegance of our experimental study and accurately captured the core findings on partially reversible energy exchange and energy recuperation within a viscoelastic bath.

Below we address point-by-point all comments and questions.

Technical comments:

#1. On page 2, 2nd paragraph after Eq.(2): “As the optical force acts opposite to the trap motion, work is performed and $\langle W(t) \rangle$ also increases.”

In 1, it seems that the tweezer force acts to the right, i.e. to the same direction as the trap moves. Please check.

Indeed the reviewer is obviously right, and it is the friction which acts against the trap motion. We have updated the manuscript accordingly.

#2. On all the figures no red curve are shown and only orange ones are presents. These orange curves are often revered to as the “red” ones throughout the manuscript. Please correct.

Curves in Fig 1 b and c are supposed to be red and thus referred to accordingly. From the referee’s comment, we understand that this distinction between orange and red is not as clear as we thought. We have thus updated the figures and simplified the color scheme.

#3. Another suggestion to figures: when printed on greyscale printer, it is very hard to distinguish different curves, and solid circles.

We have taken the referees suggestion into account and updated our color scheme to make it better suited to greyscale printer.

We now always have blue (dark) for work, red (intermediate) for E, and yellow (bright) for V.

#4. On page 4, before “Microscopic model”, the following sentence is not clear: “Before discussing the physical origin of ER, it is important to note that $\langle V(t) \rangle$ and $\langle W(t) \rangle$ exhibit trends similar to $\langle E(t) \rangle$ during step 3 . However, this behavior is also observed in viscous baths and can be attributed to intra-trap particle motion throughout the protocol?”

Namely, in Fig.4a the orange curve $\langle E(t) \rangle$ exhibits a plateau at the beginning of step “3”, while the blue $\langle W(t) \rangle$ curve has a local minimum. This behaviour disagrees with the message of the italic text above. Please, clarify.

By this statement, we mean that a decrease of work is also observed in viscous media, because some of the potential energy (which exists due to the presence of the trap) can be converted back into work. This is why in Fig.4a, we observe a non-monotonic behaviour for $\langle W \rangle$ (as in the non-Markovian case), while $\langle E \rangle$ (which is pure heat in this case) monotonically increases.

We have updated the manuscript to make this specific sentence more clear.

#5. Page 4, “Optimization of energy recuperation”: “ER decreases with increasing t_{neq} and approaches zero when $t_{\text{neq}} \gg t_b$.”

In Fig. 2a, ER reached zero for $t_{\text{neq}} \approx 3 t_b$, and not $\gg t_b$. Please, correct.

In this specific case, for the experiments there is still a non-negligible amount of ER for $t_{\text{neq}} = 4t_b$, while as the referee correctly pointed out, for the simulations ER reaches zero for $t_{\text{neq}} \sim 3t_b$.

We have updated the manuscript and softened our original statement to better align with the numerical findings.

#6. Page 5: “As a measure of the fraction of irreversibly lost energy (heat) during a compression/expansion cycle, we define the efficiency measure $\Sigma = W_{\text{neq}}/W_{\text{eq}}, \dots$ ”

Should not Σ be defined in terms of $\Delta W_{\text{neq}}/\Delta W_{\text{eq}}$?

The referee is perfectly right, we forgot the Δ in this expression.
We corrected this mistake in the manuscript.

#7. Regarding Fig3, please comment on simulation results for Newtonian fluids at small τ_{eq} , which appears to be more efficient than the viscoelastic bath. Are such parameter for Newton fluid even realistic?

The parameters we use here correspond to a fluid which would have the same zero-shear friction as our viscoelastic fluid, which is around 0.3 Pa.s. This is typically similar to some motor oil which are used in the industry. We could reach such viscosity by using a higher fraction of glycerol in our glycerol-water mixture (~90% needed), but the refraction index of the resulting fluid becomes too close to silica, making it impossible to trap our particle. We have updated the Methods section and now mention why we do not use such a high volume fraction of glycerol for our experiments in a Newtonian fluid.

#8. On Fig .4 a and b , the vertical axis has different origin, please clarify.

Indeed, for Fig 4b, we shifted the x and y axis so that all quantities (time and E) are taken as zero at the start of step 3. We added a note to clarify this point.

#9. Eq.5, in the r.h.s. not $\text{grad } U$, but $\text{grad } V$.

Although $U = V$ in the Newtonian case, we agree with the referee that we should use the same notations as in Eq.3. We have updated the manuscript accordingly.

#10. Page 10, section “Extracting the parameters of the microscopic model”: “We run Langevin simulations of the full cycle using Eqs. (5) and (6)”, May be Eqs. (3) and (4)?

Indeed, this is a typo, which we have corrected accordingly.

Reviewer #5

The manuscript presents the findings of carefully conducted experiments in complex fluids. The authors measure how much of the energy stored by the elasticity of a viscoelastic fluid can be returned to a tracer particle and discuss under what conditions this can be maximised. I find their focus on the system's thermodynamics (as opposed to classic microrheology studies) novel, fresh, and relevant for microscale transport and thermodynamics. This is a solid, timely and relevant manuscript. However, in view of the relevance for the stochastic thermodynamics community, I find that some concepts should be discussed with higher care and the connection with some of the existing literature on microrheology should be made clearer. If the authors manage to successfully address these points and the ones I make below, I will be happy to recommend their manuscript for publication. In the following, you will find my detailed comments on the manuscript and a discussion of the authors' responses to the first round of review.

We thank the reviewer for their thoughtful and constructive report, and we are pleased that they find our work both novel and relevant to the field. We appreciate their recognition of our focus on the thermodynamic aspects of microscale transport in complex fluids.

We fully acknowledge the need for greater clarity and precision in the presentation of some of the core concepts introduced in the manuscript, particularly in light of their relevance to the stochastic thermodynamics community. We also appreciate the recommendation to better contextualize our findings within the broader microrheology literature.

In the revised version of the manuscript, we have implemented many changes aimed at improving the general clarity and reinforcing the connection with existing studies. Below, we address each of the reviewer's comments in detail and outline the corresponding modifications made to the manuscript.

MAIN COMMENTS

1. Given the relevance of the results of this manuscript for stochastic thermodynamics, it is necessary to expand the discussion on the definition of E , heat, what is a bath and the fluid, also following the comment of reviewer #2, which has not been fully addressed by the authors.

This is a difficult discussion because of the additional subtleties introduced by the viscoelastic nature of the fluid under consideration. Below, I will summarise my thoughts on the issue, which the authors may consider in their extended discussion. What is clear in the experimental situation under consideration is the physical distinction between the tracer and the fluid. The identification of the fluid with a bath can be more problematic. As remarked by reviewer #2, usually, a bath is, by definition, always at equilibrium, while in these experiments, the elastic component of the fluid can be displaced from equilibrium. I am not sure this warrants the name of nonequilibrium bath, which is usually reserved for baths that are continuously driven out of equilibrium (as, for example, bacterial baths).

Similarly, heat is usually defined as the energy exchanged with the uncontrollable degrees of freedom that make up the bath, as nicely put in Sekimoto, K.: Heat Viewed at Different Scales. Lect. Notes Phys. 799, 203–220 (2010) DOI 10.1007/978-3-642-05411-2_6 "Roughly speaking, heat is the energy exchanged with or among the degrees of freedom that do not

emerge in explicit observation and description. Once we fix the level of description, for example, of the Langevin equation or of the master equation, we retain certain degrees of freedom and eliminate other degrees of freedom from the evolution equation. Then heat is the work done by the retained degrees of freedom against the thermal environment that represents the eliminated degrees of freedom.” This would suggest that if the fluid was to be interpreted as a bath, E would have to be interpreted as heat. However, this would be incorrect because part of E can be recuperated. I agree with the authors’ identification of an elastic energy contribution provided by the fluid. Perhaps it would be more precise to name it U_{elastic} . One may be tempted to suggest that the viscous properties of the fluid represent a proper bath, and its elastic properties, while being related to the fluid, do not.

We fully agree with the referee’s discussion, as we also initially directly interpreted E as heat. We eventually opted to introduce E , to avoid the notion of negative heat.

We also chose the U_{bath} naming, in regards to the generic aspect of this source of internal energy: it doesn’t need to originate from an elastic response. For instance, numerical simulations have recently shown that similar effects can arise within a hard-sphere glass (see Ditz et al *Physical Review E* 110.5 (2024): 054603). In this situation the excess of internal energy inside the bath has an entropic origin.

Splitting the bath in two is exactly what we did in our model, with the tracer particle corresponding to an ideal viscous bath and the bath particle being associated with the viscoelastic properties.

Following the referee’s comment we have made several changes to the manuscript.

First we reworked the paragraph in the model section, where we now discuss what is included in the system and what is coarse-grained into the bath. We now explicitly mention that the resulting non-Markovian bath is not an ideal bath anymore. We now also avoid the term “nonequilibrium bath” but say instead that the bath can be “temporarily driven out of equilibrium”

Second, we added a sentence discussing the possibility of different storage mechanisms (outside of an elastic network) for non-Markovian baths, to justify the use of U_{bath} .

Third, we added (see below) an additional subsection in the methods “Stochastic observables in the bath particle model”, at the end of which we discuss how the heat ends up splitted between a viscous and an elastic component.

2. The microscopic model proposed in Eqs. (3) and (4) allows the description of how the potentially retrievable energy (U_{bath}) can also be dissipated as heat. It may be helpful to discuss the stochastic thermodynamics of Eqs. (3) and (4). Following Sekimoto’s definitions, one can identify the heat dissipated by the tracer into the viscous environment as

$$\delta Q_p = (\gamma \dot{x} - \xi) \circ dx$$

and the one dissipated by the elastic component of the fluid as

$$\delta Q_b = (\gamma_b \dot{x}_b - \xi_b) \circ dx_b .$$

Their sum gives the total heat $\delta Q = \delta Q_p + \delta Q_b$, which is defined without the need of the microscopic model in Eq.(2), as

$$\delta Q = \delta W - dV - dU_{\text{bath}} ,$$

as one can see by noticing that

$$\delta W - dV = - \partial V / \partial x dx$$

and that

$$dU_{\text{bath}} = \partial U_{\text{bath}} / \partial x dx + \partial U_{\text{bath}} / \partial x_b dx_b .$$

Using the model, we can indeed directly obtain the full dissipated heat $Q = Q_p + Q_b$, either by using the first law: $Q = W - V - U_{\text{bath}}$ or by deriving the heat dissipated by the bath particle itself. This is naturally confirmed in the simulations.

Unfortunately, without the model, evaluating U_{bath} is impossible (we don't have a value for x_b), which is why we decided to introduce $\delta E = \delta Q + dU_{\text{bath}}$, as this is the only observable we have access to in this situation.

We have added a new subsection "Stochastic observables in the bath particle model" in the Methods, which discusses the above derivation using Sekimoto's formalism.

3. The authors state that the average heat always has to increase. However, this is not correct for a non-stationary process (as the ones studied here) where the changes in the system's entropy may compensate for this.

The reviewer is correct that a decrease in the system's average entropy can lead to negative heat flow, even when considering ensemble averages. However, in our case, since we use a harmonic potential, the external driving only increases the system's potential energy and does not affect its entropy. Therefore, the average heat in our study must always increase.

We have expanded our statement to clarify that the system's entropy remains conserved during the driving, and therefore does not affect heat production.

4. I find the definition of energy recuperation arbitrary. Why is it taken as the difference between the local maximum and the local minimum of E ? This quantity contains both a heat contribution and an energy transfer from the fluid back to the particle.

Only the energy stored in U_{bath} should be available for recuperation, as the other one is lost as heat. Wouldn't it be clearer to define it from when the first protocol stops (i.e. when U_{bath} stops increasing)?

We fully agree that the definition of ER is imperfect, in the sense that it contains both a (positive) contribution from the dissipated heat (coming both from V and U_{bath}), and a (negative) contribution from the energy transfer from the bath to the particle (only coming from U_{bath}). As a result we typically underestimate the recuperated energy, the ER amplitude we measure is systematically smaller than the amplitude of energy transferred from the bath to the particle. As discussed above, we cannot differentiate between these two sources of internal energy, and we had to settle with this imperfect definition.

By measuring ER only during step 3, we avoid incorporating an additional heat exchange which necessarily happens during step 2, which would lead to an even larger underestimation of the energy truly recovered from the bath.

We have added additional comments in the manuscript, which discuss the presence of heat which decreases the measured energy recuperation.

5. It is not immediately clear that Eqs. (3) and (4) can be combined into a generalised Langevin equation. Reference [22] does not show this clearly. These equations are crucial for the thermodynamics description and should, therefore, be introduced and discussed with care. An important aspect to discuss is how this expression satisfies the Fluctuation

Dissipation Relation. Also, the connection with G' and G'' in microrheology should be spelt out, following the suggestions of reviewer #4.

The combination of Eqs.(3) and (4) corresponds to the formulation presented in Appendix E (p.10) of Ref.[22]. In this framework, each additional bath particle contributes an independent exponential term to the memory kernel Γ (see Eq.(E2)) and to the noise correlation function (Eq.(E3)), effectively adding a new timescale to the system's relaxation dynamics. Furthermore, when adopting the generalized Langevin equation (GLE) formalism, the connection between the memory kernel Γ and the viscoelastic moduli $G'(w)$ and $G''(w)$ in the frequency domain is addressed in Ref. [47].

To clarify these points, we have extended the discussion in the manuscript by adding two dedicated subsections in the Methods section—one detailing the structure of the memory kernel and noise, and the other explicitly discussing the correspondence with the rheological response functions.

6. I find Fig. 2b a bit too qualitative, maybe a scatter plot would be better to show the trajectory-wise relation. I also suspect that there may be other quantities which would be more appropriate for describing the “charging” of the elastic energy of the fluid and its partial recuperation.

We initially considered showing ΔE_{neq} vs ΔE_{eq} , as this would directly reflect the trajectory-wise energy exchange between the particle and the bath during steps 1 and 3. A typical heat map illustrating this relationship is shown below.

However, we ultimately chose to focus on the ER amplitude instead. Compared to ΔE_{neq} , ER is less influenced by conventional heat generation, making it a more selective indicator of reversible energy exchange with the bath.

Furthermore, by binning the data into curves, we are able to more clearly visualize how this correlation decays as a function t_{neq} , and eventually vanishes when the bath has reached equilibrium again.

We believe that the curves shown in Fig.2b offer a good compromise between clarity and interpretability. Nevertheless, we would be happy to replace them with a heat map if the referee considers it more appropriate or informative.

7. The authors claim that friction in the neq step is reduced. Why is the particle not moving faster in this step, then? Is it because the force is smaller?

We claim that the effective friction is reduced during step 3, as the particle travels the same distance as in step 1 but at a much lower—or even negative—work cost. As shown in Fig.1b, the optical force in step 1 is consistently positive, with the trap positioned ahead of the particle and actively pulling it forward. In contrast, during step 3, the force is mostly negative (except near the end, $t > 6\text{s}$), indicating that the trap trails behind while the particle is propelled forward—primarily by the relaxation of the surrounding fluid. As a result, the average force exerted by the trap in step 3 is much smaller than in step 1, despite the comparable displacement, reflecting the contribution of energy recuperation in reducing dissipation.

We have rewrote the corresponding paragraph to make our point clearer.

8. It is not entirely clear to me what the authors want to quantify by the quantity Σ . I do not find it a good illustration of energy recuperation since the fact that W_{neq} is lower than V_{neq} could also happen for a Markovian bath. Additionally, why is $-1 < \Sigma < 1$?

In a Markovian bath the first law imposes $\Delta W_{\text{neq}} > \Delta V_{\text{neq}}$ because $\Delta E_{\text{neq}} = \Delta Q_{\text{neq}} > 0$. The idea of Σ , was to directly show the fraction of work which can be recovered through the cycle, which we believe is the most relevant quantity for any applications. It also enabled us to make a direct comparison to an equivalent Markovian system, showing both a quantitative enhancement, but also a qualitative different behaviour with a non-monotonic behaviour.

We have rewritten this paragraph to remove the piston analogy and instead rely on the numbered steps framework already introduced earlier in the manuscript. Additionally, we now explicitly discuss why the energy recovery ratio Σ is bounded between -1 and 1.

9. The statement “The amount of energy stored in each channel varies with the driving time t_d , and typically, at higher trap velocities (shorter t_d)” is interesting but it should be expanded and properly justified.

We agree with the referee that this statement is too vague and should be better discussed. We have rewritten and expanded this paragraph in the revised version.

COMMENTS ON REBUTTAL

I have read the reviewers' comments and the authors' replies. I've found the comments very insightful and the authors addressed them properly except for the parts that I also included in my comments and for the cases discussed below.

As mentioned above, I agree with reviewer #2 about the bath, and clearer language and higher care in the discussion are probably needed. I do not, however, agree with reviewer #2's suggestion to drop the term fictitious for the "bath particle". This is a mathematically convenient way of treating the memory of the fluid, not a physical particle, in contrast with the tracer. Whether these degrees of freedom should be attributed to the system or the bath is an interesting question. What is clear to me is that these additional degrees of freedom are transiently displaced from equilibrium by the tracer's motion. In this process, they can exchange energy with the tracer, temporarily storing it and eventually releasing it back. In addressing this and other comments, the authors included statements about the possibility of including the bath particle within the system.

This would change the thermodynamic interpretation so the authors should be careful with such statements.

I find the piston analogy still confusing (I agree with reviewer #4)

We thank the referee for the thoughtful remarks and agree that clearer language was needed, particularly regarding the system–bath distinction. In the previous revision of the manuscript, we had already carefully revised the discussion and improved the overall phrasing to ensure conceptual clarity.

We also fully agree that the "bath particle" introduced in the model should remain part of the bath and only serve as a mathematical representation of the fluid's memory. Our previous mention of including it "within the system" was only meant to illustrate that, in theoretical models, one can sometimes describe non-Markovian dynamics by extending the system to include the auxiliary (hidden) variables. Such a Markovian embedding is purely a modeling convenience and is obviously impossible for the experimental data. It does not imply a reinterpretation of the actual experiments.

Regarding the piston analogy, we thought it would help to make a connection between our system and a situation well-known to the readers. We have therefore expanded and clarified this section with reads now:

"This process bears some resemblance with the operation of a classical heat engine, where a piston moves within a cylinder containing a compressible working gas. In such engines, energy is cyclically injected and extracted through the compression and expansion of the gas, with part of the input work stored temporarily as internal energy and the rest dissipated as heat due to friction and other losses. Similarly, in our system, the colloidal particle (piston) is periodically driven within a harmonic trap (cylinder) immersed in a viscoelastic fluid (working medium). During forward motion, the particle squeezes the viscoelastic network, analogous to compressing a gas, thereby injecting energy into both the trap and the surrounding medium. Due to the fluid's delayed response, a fraction of this energy is stored elastically in the bath's internal degrees of freedom rather than being immediately dissipated. Upon reversing the trap motion, part of the stored

energy is released and can be reinjected into the system, analogous to the expansion stroke of the piston where stored energy contributes to useful output. However, just as in macroscopic engines, not all energy is recovered; viscous friction leads to irreversible heat dissipation, thereby limiting the efficiency of the cycle.”

In case, the referee still finds such a connection confusing, this paragraph can be easily omitted without disturbing the flow of the paper.

MINOR COMMENTS

- Why isn't E plotted in Fig. 1d?

We thought Fig. 1d was already very crowded, and thus decided to not show $E = U_{\text{bath}} + Q$ because it essentially perfectly matches the experiments. This can be seen in Fig. 5a where we compare experiments and simulation with the same parameters as in Fig 1.

- Contrary to what is stated in the main text, Fig. 4a doesn't show Q but only E, a more precise language is needed.

Indeed, this mistake has been corrected, and all references to heat (which are equivalent in the Newtonian fluid shown in Figs 4a and b) have been removed.

- The energy given by the particle to the fluid, E, is one of the key quantities in the manuscript. Its definition probably deserves a display equation. I also find it clearer to give its operational definition in terms of the measurable quantities V and W .

We have updated the manuscript as suggested.

- I find the statement “In contrast to previous experiments, where energy transfer from a bath to a particle has been attributed to thermal fluctuations [43, 44], the key distinction here is that this energy transfer is observed on an ensemble average.” unclear.

With this statement, we intended to emphasize that temporary energy transfers from the bath to the particle are fully expected due to thermal fluctuations and are not unique to non-Markovian baths. Such fluctuations can lead to transient energy injection even in equilibrium, Markovian systems. However, what distinguishes our findings is that this energy transfer persists at the ensemble level—i.e., it remains visible after averaging over many trajectories. This implies a systematic, rather than purely stochastic, mechanism enabled by the bath's memory.

To clarify this point, we have revised the sentence in the manuscript.

- What is the living network of giant micelles?

In contrast to conventional polymers, whose dynamics are primarily governed by filament reptation, worm-like micelles possess additional degrees of freedom due to their ability to continuously break and reform. This is possible because the surfactant molecules that constitute the micelles can freely diffuse in solution. The resulting highly dynamic structure is conventionally referred to as a “living network” in the literature.

We have therefore updated the reference from “living network” to “highly dynamic network” for clarity.

- This sentence is unclear “Obviously, the energy exchange between the particle and the viscoelastic (compared to a viscous) fluid between steps 1 and 3 is fundamentally different although the trap motion was simply reversed, and the mean particle velocity \dot{x} is almost identical (Fig. 1b).”

We fully agree with the referee that this sentence is not clear enough. We have updated it accordingly.

- Are the error bars for ER and DV smaller than the symbols in Fig 2a?

Yes, for these two curves, the error bars (defined as the standard error of the mean, SEM) are consistently smaller than the symbol size. This is because the corresponding observables have relatively small amplitudes, and consequently, the SEM is also typically small.

- It could be interesting to see a plot like Fig. 3 for the case with 2 timescales

Including a second bath particle in our model significantly improves the agreement with the experimental data presented in Fig. 3, as shown in the updated comparison below. This improvement directly correlates with our observations of the general relaxation dynamics in Fig. 5b, which are more accurately captured by the inclusion of an additional viscoelastic timescale.

- In the Methods section “Optical tweezers setup” the authors say that $V(x) = \log p$. However, this is not correct, as the probability normalisation also depends on κ .

The referee is obviously correct. The probability normalization was omitted in this expression, so that we have the potential energy starting at zero: $V(0) = 0$.

We thus added a “+ cst” to the expression.

- The authors show that including an additional time scale improves the modelling of the system. How does the thermodynamic interpretation change if 2 time scales are present?

The thermodynamics interpretation doesn't fundamentally change with the addition of one (or many) additional timescales. What matters is the capacity of the bath to temporarily store and restore energy (and at different rates), through the presence of hidden degrees of freedom.

However, the presence of one additional timescale means that energy can also flow between the two associated channels. Such behavior, predicted from the model, has recently been observed experimentally in our lab and can lead to complex non-monotonic relaxation processes (see Salil et al. Phys. Rev. Res, **7**, 033084 (2025))

ANSWER TO REVIEWERS' COMMENTS

Reviewer #3 (Remarks to the Author):

The authors have responded convincingly to all my questions and addressed all my concerns. Therefore, I recommend publication of the manuscript in its present form in Nature Communications.

Reviewer #5 (Remarks to the Author):

The authors have satisfactorily addressed all my comments, and I am happy to recommend the manuscript for publication.

We thank both reviewers for their careful assessment of our work and for recommending publication. We greatly appreciate their constructive feedback throughout the review process, which has helped us to improve the clarity and quality of the manuscript.

Reviewer Report on Energy Recuperation of Driven Colloids in Non-Markovian Baths

The manuscript presents the findings of carefully conducted experiments in complex fluids. The authors measure how much of the energy stored by the elasticity of a viscoelastic fluid can be returned to a tracer particle and discuss under what conditions this can be maximised. I find their focus on the system's thermodynamics (as opposed to classic microrheology studies) novel, fresh, and relevant for microscale transport and thermodynamics. This is a solid, timely and relevant manuscript. However, in view of the relevance for the stochastic thermodynamics community, I find that some concepts should be discussed with higher care and the connection with some of the existing literature on microrheology should be made clearer. If the authors manage to successfully address these points and the ones I make below, I will be happy to recommend their manuscript for publication. In the following, you will find my detailed comments on the manuscript and a discussion of the authors' responses to the first round of review.

MAIN COMMENTS

1. Given the relevance of the results of this manuscript for stochastic thermodynamics, it is necessary to expand the discussion on the definition of E , heat, what is a bath and the fluid, also following the comment of reviewer #2, which has not been fully addressed by the authors.

This is a difficult discussion because of the additional subtleties introduced by the viscoelastic nature of the fluid under consideration. Below, I will summarise my thoughts on the issue, which the authors may consider in their extended discussion.

What is clear in the experimental situation under consideration is the physical distinction between the tracer and the fluid. The identification of the fluid with a bath can be more problematic. As remarked by reviewer #2, usually, a bath is, by definition, always at equilibrium, while in these experiments, the elastic component of the fluid can be displaced from equilibrium. I am not sure this warrants the name of nonequilibrium bath, which is usually reserved for baths that are continuously driven out of equilibrium (as, for example, bacterial baths).

Similarly, heat is usually defined as the energy exchanged with the uncontrollable degrees of freedom that make up the bath, as nicely put in Sekimoto, K.: Heat Viewed at Different Scales. Lect. Notes Phys. 799, 203–220 (2010) DOI 10.1007/978-3-642-05411-2 6 “Roughly speaking, heat is the energy exchanged with or among the degrees of freedom that do not emerge in explicit observation and description. Once we fix the level of description, for example, of the Langevin equation or of the master equation, we retain certain degrees of freedom and eliminate other degrees of freedom from the evolution equation. Then heat is the work done by the retained degrees of freedom against the thermal environment that represents the eliminated degrees of freedom.” This would suggest that if the fluid was to be interpreted as a bath, E would have to be interpreted as heat. However, this would be incorrect because part of E can be recuperated. I agree with the authors’ identification of an elastic energy contribution provided by the fluid. Perhaps it would be more precise to name it $U_{elastic}$. One may be tempted to suggest that the viscous properties of the fluid represent a proper bath, and its elastic properties, while being related to the fluid, do not.

2. The microscopic model proposed in Eqs. (3) and (4) allows the description of how the potentially retrievable energy (U_{bath}) can also be dissipated as heat. It may be helpful to discuss the stochastic thermodynamics of Eqs. (3) and (4). Following Sekimoto’s definitions, one can identify the heat dissipated by the tracer into the viscous environment as

$$\delta Q_p = (\gamma \dot{x} - \xi) \circ dx$$

and the one dissipated by the elastic component of the fluid as

$$\delta Q_b = (\gamma_b \dot{x}_b - \xi_b) \circ dx_b .$$

Their sum gives the total heat $\delta Q = \delta Q_p + \delta Q_b$, which is defined without the need of the microscopic model in Eq.(2), as

$$\delta Q = \delta W - dV - dU_{bath} ,$$

as one can see by noticing that

$$\delta W - dV = -\frac{\partial V}{\partial x} dx$$

and that

$$dU_{bath} = \frac{\partial U_{bath}}{\partial x} dx + \frac{\partial U_{bath}}{\partial x_b} dx_b .$$

3. The authors state that the average heat always has to increase. However, this is not correct for a non-stationary process (as the ones studied here) where the changes in the system’s entropy may compensate for this.

4. I find the definition of energy recuperation arbitrary. Why is it taken as the difference between the local maximum and the local minimum of E ? This quantity contains both a heat contribution and an energy transfer from the fluid back to the particle. Only the energy stored in U_{bath} should be available for recuperation, as the other one is lost as heat. Wouldn't it be clearer to define it from when the first protocol stops (i.e. when U_{bath} stops increasing)?
5. It is not immediately clear that Eqs. (3) and (4) can be combined into a generalised Langevin equation. Reference [22] does not show this clearly. These equations are crucial for the thermodynamics description and should, therefore, be introduced and discussed with care. An important aspect to discuss is how this expression satisfies the Fluctuation Dissipation Relation. Also, the connection with G' and G'' in microrheology should be spelt out, following the suggestions of reviewer #4.
6. I find Fig. 2b a bit too qualitative, maybe a scatter plot would be better to show the trajectory-wise relation. I also suspect that there may be other quantities which would be more appropriate for describing the "charging" of the elastic energy of the fluid and its partial recuperation.
7. The authors claim that friction in the neq step is reduced. Why is the particle not moving faster in this step, then? Is it because the force is smaller?
8. It is not entirely clear to me what the authors want to quantify by the quantity Σ . I do not find it a good illustration of energy recuperation since the fact that W_{neq} is lower than V_{neq} could also happen for a Markovian bath. Additionally, why is $-1 < \Sigma < 1$?
9. The statement "The amount of energy stored in each channel varies with the driving time t_d , and typically, at higher trap velocities (shorter t_d)" is interesting but it should be expanded and properly justified.

COMMENTS ON REBUTTAL

I have read the reviewers' comments and the authors' replies. I've found the comments very insightful and the authors addressed them properly except for the parts that I also included in my comments and for the cases discussed below.

As mentioned above, I agree with reviewer #2 about the bath, and clearer language and higher care in the discussion are probably needed. I do not, however, agree with reviewer #2's suggestion to drop the term fictitious for the "bath particle". This is a mathematically convenient way of treating the memory of the fluid, not a physical particle, in contrast with the tracer. Whether these degrees of freedom should be attributed to the system or the bath is an interesting question. What is clear to me is that these

additional degrees of freedom are transiently displaced from equilibrium by the tracer’s motion. In this process, they can exchange energy with the tracer, temporarily storing it and eventually releasing it back. In addressing this and other comments, the authors included statements about the possibility of including the bath particle within the system. This would change the thermodynamic interpretation so the authors should be careful with such statements.

I find the piston analogy still confusing (I agree with reviewer #4)

MINOR COMMENTS

- Why isn’t E plotted in Fig. 1d?
- Contrary to what is stated in the main text, Fig. 4a doesn’t show Q but only E , a more precise language is needed.
- The energy given by the particle to the fluid, E , is one of the key quantities in the manuscript. Its definition probably deserves a display equation. I also find it clearer to give its operational definition in terms of the measurable quantities V and W .
- I find the statement “In contrast to previous experiments, where energy transfer from a bath to a particle has been attributed to thermal fluctuations [43, 44], the key distinction here is that this energy transfer is observed on an ensemble average.” unclear.
- What is the living network of giant micelles?
- This sentence is unclear “Obviously, the energy exchange between the particle and the viscoelastic (compared to a viscous) fluid between steps 1 and 3 is fundamentally different although the trap motion was simply reversed, and the mean particle velocity $\langle \dot{x} \rangle$ is almost identical (Fig. 1b).”
- Are the error bars for ER and DV smaller than the symbols in Fig 2a?
- It could be interesting to see a plot like Fig. 3 for the case with 2 timescales
- In the Methods section “Optical tweezers setup” the authors say that $V(x) = \log p$. However, this is not correct, as the probability normalisation also depends on κ .
- The authors show that including an additional time scale improves the modelling of the system. How does the thermodynamic interpretation change if 2 time scales are present?